# Multi-year presence of humpback whales in the Atlantic sector of the Southern Ocean but not during El Niño

Elena Schall [1✉], Karolin Thomisch[1], Olaf Boebel[1], Gabriele Gerlach [2,3], Sari Mangia Woods[1,4], Ahmed El-Gabbas [1] & Ilse Van Opzeeland[1,2]

Humpback whales are thought to undertake annual migrations between their low latitude breeding grounds and high latitude feeding grounds. However, under specific conditions, humpback whales sometimes change their migratory destination or skip migration overall. Here we document the surprising persistent presence of humpback whales in the Atlantic sector of the Southern Ocean during five years (2011, 2012, 2013, 2017, and 2018) using passive acoustic data. However, in the El Niño years 2015 and 2016, humpback whales were virtually absent. Our data show that humpback whales are systematically present in the Atlantic sector of the Southern Ocean and suggest that these whales are particularly sensitive to climate oscillations which have profound effects on winds, sea ice extent, primary production, and especially krill productivity.

[1] Alfred Wegener Institute for Polar and Marine Research, Bremerhaven, Germany. [2] Helmholtz Institute for Functional Marine Biodiversity, Carl von Ossietzky University Oldenburg, Oldenburg, Germany. [3] Carl von Ossietzky University Oldenburg, Oldenburg, Germany. [4] Marine Evolution and Conservation, Groningen Institute of Evolutionary Life Sciences, University of Groningen, Groningen, The Netherlands. ✉email: elena.schall@awi.de

Humpback whales (*Megaptera novaeangliae*) inhabit all major oceans but these iconic large predators were thought to extend their range to polar and subpolar ecosystems only to feed during the summer months[1]. To reach these high latitude productive feeding areas, humpback whales undertake one of the longest mammalian migrations[2]. In the Atlantic sector of the Southern Ocean (ASSO), the investigation of humpback whale distribution by ship-based sighting surveys is only feasible during the austral summer and still limited due to necessary logistic effort. Therefore, systematic data on their (year-round) presence, abundance, and spatial distribution are missing for the ASSO. Insights on distribution are however vital for understanding their present and future role as large predators in structuring the Southern Ocean ecosystem[3,4]. A long-term autonomous passive acoustic monitoring (PAM) network was installed in 2010 to record humpback whales in their natural Antarctic environment year-round. Humpback whales are excellent candidates for PAM studies due to their year-round vocal activity of all sex and age classes[5–7]. To improve the understanding of the ecological conditions under which humpback whales use the area as a feeding ground, we investigated the inter-annual changes in humpback whale acoustic presence in relation to three environmental parameters that are key to the Southern Ocean: (1) The Southern Annular Mode (SAM) which is the dominant pattern of natural climate variability in polar and subpolar regions of the Southern Hemisphere. (2) The El Niño Southern Oscillation (ENSO) causes periodic fluctuation of sea surface temperature and air pressure originating from the tropical Pacific. Both climate oscillations have large effects on the Southern Ocean productivity[8–11]. (3) Local sea ice concentration (SIC) directly affects whale access to open water areas which is necessary for breathing. Indirect effects of local sea ice concentration entail its impact on the distribution of primary productivity, which in turn drives the distribution of Antarctic krill (*Euphausia superba*), the humpback whale's key prey species[7,8,12,13]. The large-scale mooring network in the ASSO that we have been maintaining for more than ten years[14] allowed us to relate the long-term trend in humpback whale acoustic presence to long-term trends in SIC and climate oscillations.

## Results and discussion

**Perennial humpback whale acoustic presence.** We analyzed passive acoustic data of five recording positions (G1–G5) on the Greenwich Meridian from a mooring network throughout the ASSO from December 2010 to September 2018 (Fig. 1).

At the four oceanic recording locations (G1–G4), humpback whales were acoustically present during summer and autumn of the years 2011, 2012, 2013, 2017, and 2018 (i.e., times at which data were available for each recording position), coinciding with periods of low SIC (Fig. 2a, Supplementary Note 2, Supplementary Note 3). The high proportion of hours with humpback whale acoustic presence during autumn at G1–G4 coincided with the known timing of onset of singing behavior in Southern Hemisphere humpback whale males in lower latitude waters[15,16]. During this period, two or more individuals often were vocalizing at the same time in our recordings and acoustic activity was registered at all four locations in parallel, indicating the presence of multiple animals. Close to the coastal recording location (G5) where high sea ice concentrations were common during most months, humpback whales were acoustically absent or appeared only at low rates (e.g., during the years 2011–2013; Supplementary Note 2, Supplementary Note 3). At the same position (G5) and one of the oceanic locations (G3), humpback whales were acoustically present also during winter months, when SIC reached almost 100% (Fig. 2a, Supplementary Note 2, Supplementary Note 3). Although humpback whale winter acoustic presence was limited compared to the summer months, the occurrence of calls in winter was persistent between years occurring at multiple sites (Supplementary Note 3).

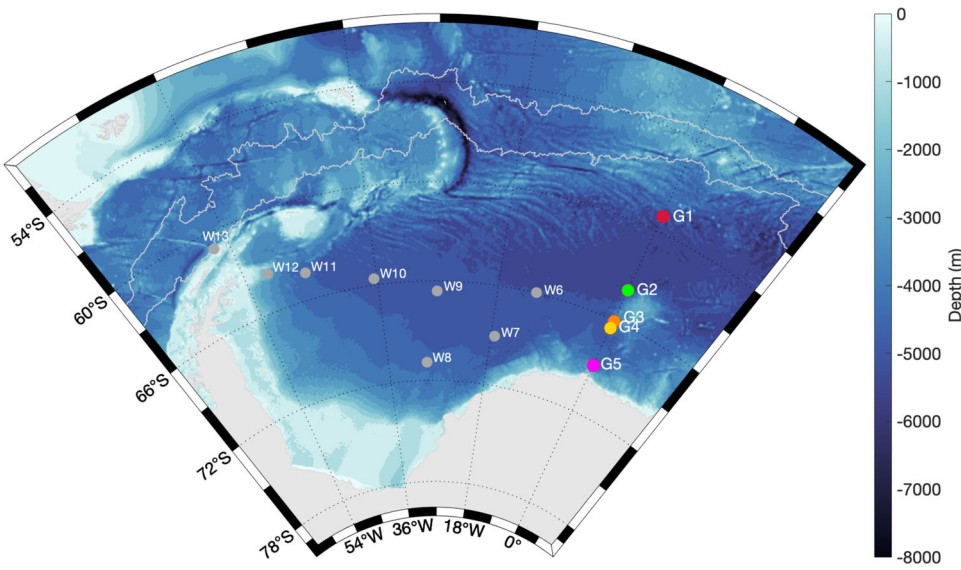

**Fig. 1 The Atlantic sector of the Southern Ocean and recording positions on the Greenwich Meridian.** Bathymetric map of the Atlantic sector of the Southern Ocean (ASSO) including the geographic positions of the HAFOS (Hybrid Antarctic Float Observation System) mooring network in the ASSO (coastline and bathymetry data were obtained from ref. [61, 62]). The five mooring positions, G1–G5, visualized with colored dots (i.e., red, green, orange, yellow, and magenta), represent the recording locations of the receivers (moored between 2010 and 2018) which were analyzed during this study. Positions G1–G5 form part of the HAFOS long-term mooring network (gray dots[14]). The other recording positions (W6–13) were only active during 2013 and were, therefore, not included here (but see ref. [25] for details). Light gray lines represent the minimum and maximum of the annual wintertime (21 June–21 September) maximum sea ice extent during the study period (2011–2018)[63]. Please note, that the lines shown do not delineate the sea ice extent of the specific years with the maximum and minimum wintertime maximum sea ice extent, but - calculated independently for each longitude—the multi-year composite of the maximum and minimum of the wintertime maximum sea ice extent during this period.

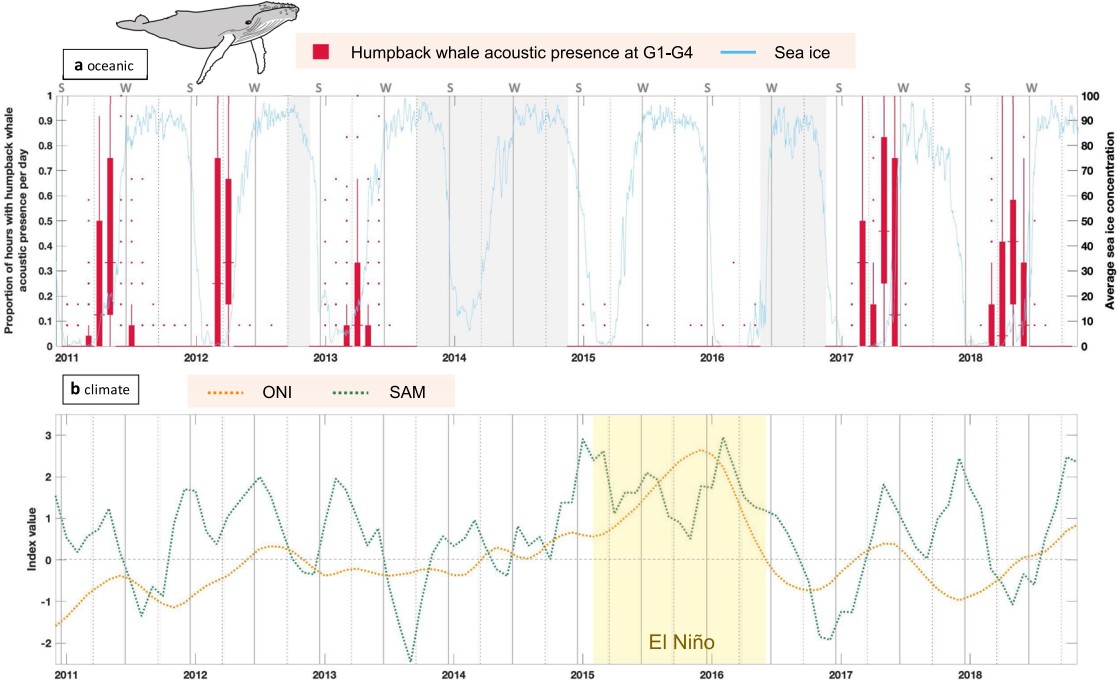

**Fig. 2 Perennial humpback whale acoustic presence in relation to sea ice concentration, SAM and ONI. a** Boxplot of proportion of hours per day with humpback whale acoustic presence for each month from the four oceanic recording locations (G1–G4) on the Greenwich Meridian from December 2010 until September 2018 (red bars; $n = 4614$ days of recordings). Red dots represent outliers (on a discrete scale as proportion of hours per day). Gray-shaded areas represent months without recording data. The blue solid line and the right $y$-axis depict the daily averaged sea ice concentration within a 50 km radius around recording locations. **b** Climatic variations from 2011 until 2018 indicated by 3-month running means of the Southern Annular Mode index (SAM) as a dominant pattern of natural climate variability in polar and subpolar regions of the Southern Hemisphere and the Oceanic Niño Index (ONI) representing the periodic fluctuation of sea surface temperature and air pressure originating from the tropical Pacific. Time span of strong El Niño phase in 2015/16 is indicated by the yellow rectangle. Vertical gray lines indicate the onset of summer (S) and winter (W) and vertical dotted lines indicate the onset of spring and autumn (based on equinoxes and solstices). Horizontal dashed line represents zero-orientation line.

**Humpback whale acoustic absence during El Niño.** In contrast to previous and following years, only very low numbers of humpback whale vocalizations were detected in 2015 and 2016 at all locations (Fig. 2a, Supplementary Note 2, Supplementary Note 3), whereas vocalizations of other species, e.g., the Antarctic minke whale (*Balaenoptera bonaerensis*), were detected during this time (see ref. [17]), excluding a technical artifact. Our findings are supported by the absence of opportunistic visual observations of humpback whales in the ASSO during the summer seasons 2014/2015 and 2015/2016 and only a few sightings during systematic ship-based and aerial surveys during January 2015[18–21]. During the same years, 2015 and 2016, when humpback whales were virtually absent, SAM and ENSO (represented by the Oceanic Niño Index, ONI) both simultaneously were in strong positive phases and one of the strongest El Niño phases since the beginning of measurements was registered[22] (Fig. 2b).

Modeling the effect of SIC, SAM, and ONI on the acoustic presence of humpback whales at the study location revealed that mainly SIC and ONI explain the observed pattern of humpback whale acoustic presence in the ASSO. The smoothed effects of month and SIC were highly significant because these variables explain seasonality in humpback whale presence on the feeding ground (Table 1, Fig. 3;[7,12]) in 5 out of 7 years. The model showed that ONI in the positive phase predicts a significantly lower probability of humpback whale acoustic presence than ONI in neutral or negative phases (Table 1, Fig. 3). The smoothed effect of the SAM index was not statistically significant (Table 1, Fig. 3). The model prediction for the SAM index showed lower predicted values at negative and high positive index values, although with higher uncertainties (Fig. 3). This appears

reasonable when looking at the original time-series. The negative phases of SAM were usually registered during winter when acoustic presences are naturally low, and extreme positive phases were only registered during summer 2015 and 2016 (Fig. 2). Uncertainties at extreme index values (also for ONI) are high because these values are rare in the analyzed time-series, which potentially also explains the resulting non-significant effect of SAM. To quantify the relationship between humpback whale

---

**Table 1 Results of best-fit model.**

**Formula: PA ~ s(SIC) + ONI + s(SAM) + s(Month)**

**Parametric coefficients:**

| | Estimate | Std. error | t value | Pr(>\|t\|) |
|---|---|---|---|---|
| ONI $_{Positive}$ | − 5.8490 | 0.9888 | − 5.915 | 3.87e−09*** |
| ONI $_{Positive}$ − ONI$_{Neutral}$ | 3.6750 | 0.9682 | 3.796 | 0.000151*** |
| ONI $_{Positive}$ − ONI$_{Negative}$ | 3.7672 | 1.0207 | 3.691 | 0.000229*** |

**Approximate significance of smooth terms:**

| | edf | Ref.df | F | p-value |
|---|---|---|---|---|
| s(SIC) | 3.381 | 3.381 | 9.576 | 1.34e−06 *** |
| s(SAM) | 2.103 | 2.103 | 1.561 | 0.167 |
| s(Month) | 4.635 | 8.000 | 4.387 | 7.23e−07 *** |

R-sq.(adj) = 0.485.

Summary of the best-fit model for the acoustic presence of humpback whales at stations G3/G4, including SIC, SAM, and month as smooth terms, as well as ONI as a categorical predictor ($n = 2629$ days of recordings). Note that the factor levels of ONI as a categorical predictor are listed under the parametric coefficients. Segment headings are highlighted in bold.

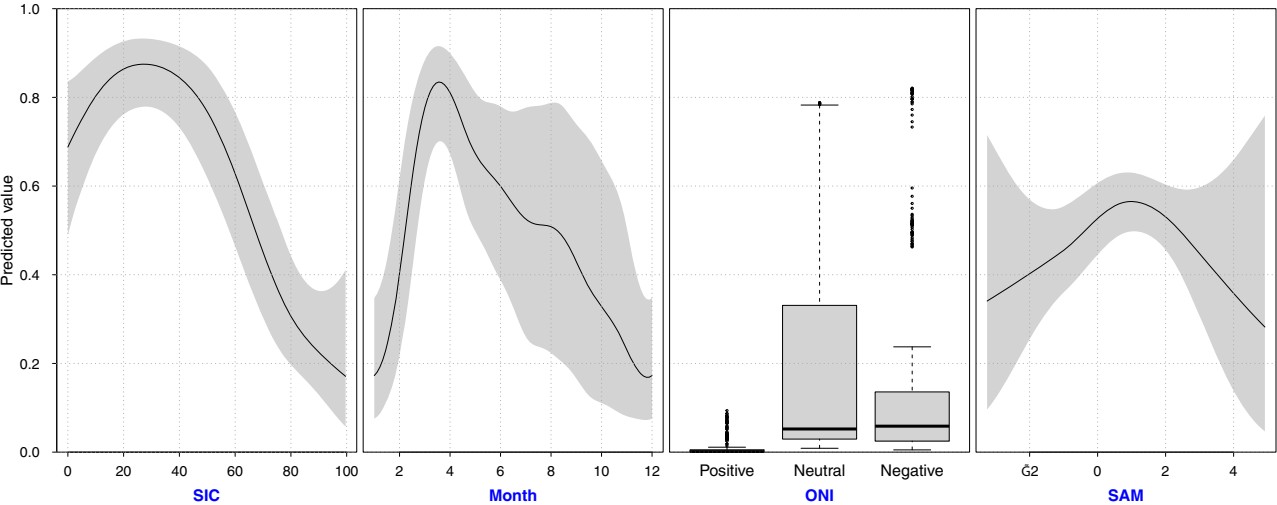

**Fig. 3 Humpback whale acoustic presence modeled as a function of SIC, SAM, and ONI.** Model predictions of the best-fit model for the acoustic presence of humpback whales at stations G3/G4, including the smooth terms SIC, month, and SAM as well as ONI as a categorical predictor (see "Methods" for further explanation of categories). Gray-shaded areas in line plots depict 95%-confidence intervals ($n = 2629$ days of recordings).

presence and climate indices with higher certainty, much longer time-series than presented here would be required.

**Ecological pathways from environmental variabilities to humpback whales.** Local sea ice concentration is one of the most important environmental factors explaining the spatio-temporal distribution of baleen whales in the Southern Ocean[12,17,23,24]. Similarly, sea ice dynamics play an important role in the intra-annual timing of humpback whale presence in the ASSO, showing that they move out of the area when SIC increases and that humpback whales are rarely present at SIC > 80% (see also refs. [7,25]). Additionally, our data indicate that large-scale climate variabilities drive the major inter-annual variability in the acoustic presence of humpback whales on a Southern Ocean feeding ground. The most likely pathway by which climate variabilities such as ENSO and SAM could affect humpback whale presence in the Southern Ocean is through their influence on Antarctic krill (*Euphausia superba*)[11], since the availability and distribution of this primary prey species most likely is the main driver behind the spatio-temporal distribution of humpback whales in the ASSO.

Both SAM and ENSO are factors influencing the spatial distribution and biomass of Antarctic krill by affecting winds, cloud cover, currents, sea surface temperature, and sea ice extent[8–10,26–29]. During the positive phase of SAM, the westerly wind belt surrounding the Antarctic continent contracts toward the continental shelf and climatic conditions north of the wind belt change to warmer, windier, and cloudier weather. During positive values of SAM, the oceanic feeding areas of humpback whales in the ASSO show signs of reduced sea ice extent, reduced primary production, and, in turn, also reduced krill densities[8,9]. Especially north of the Antarctic Polar Front, sea surface temperature increases and chlorophyll concentration decreases during positive SAM[30,31]. ENSO has the strongest effects on the Pacific sector of the Southern Ocean, including the Western Antarctic Peninsula[8,32]. The Western Antarctic Peninsula is a key habitat for Antarctic krill as a spawning and nursery ground, from which krill is transported with the Antarctic Circumpolar Current (ACC) 'conveyor belt' into north-eastern areas of the ASSO[8]. During or after the positive phase of ENSO, El Niño caused warming and the reduction of sea ice at the Western Antarctic Peninsula. Due to these climate conditions, less krill might be recruited from the Antarctic Peninsula toward the

oceanic regions of the ASSO, although this process is probably delayed by multiple months considering the estimated transport speed of the ACC[8,10,11,33]. Also, during years of El Niño, a manifestation of negative sea surface temperature anomalies in the southwest Atlantic, termed the Antarctic dipole, is common which probably affects productivity in this area[10,11,30]. We, therefore, hypothesize that during the years 2015 and 2016, positive phases of both SAM and ENSO led to reduced densities of krill on the oceanic feeding grounds of humpback whales in the ASSO while potentially creating alternative prey resources in other areas.

Among baleen whale species, the humpback whale is probably the most flexible when it comes to ecological requirements, being able to adapt to wide-ranging changes in the environment with alternative migration and feeding strategies[34,35]. This adaptivity is probably also the reason for the continued presence of at least some humpback whales in the ASSO during austral winter. Especially female and juvenile humpback whales tend to prolong their stay on the feeding grounds or even skip migration entirely in order to fuel growth, pregnancy, or lactation with additional winter feeding[32,36]. During 2015 and 2016, the main part of the South Atlantic humpback whales (probably individuals from breeding stocks from South America and Africa[37]) might have adapted their migration routes to exploit areas of high productivity elsewhere than in the ASSO[34]. For example, an unusual assemblage of humpback whale 'super groups' was documented in 2015 and 2016 in the southern Benguela upwelling system off South Africa[38,39]. Furthermore, in 2016, exceptional recordings of humpback whale song were made close to the west coast of South Africa[39,40] which indicates not only the displacement of the feeding area but also multifaceted habitat use (i.e., also including reproductive activities) along this displacement. Humpback whales acoustically and physically absent from the ASSO during 2015 and 2016 might have been exploiting alternative habitats and prey resources along the west coast of South Africa[34,38] or other yet undiscovered high productivity areas in the South Atlantic or adjacent waters. To date, the current knowledge on spatio-temporal trends in productivity hotspots in the Southern Hemisphere is nevertheless too sparse to explain trends in migratory predator distribution with certainty. In this context, the maintenance and implementation of further long-term observation systems such as the HAFOS mooring network from which the analyzed recordings originate

(see Fig. 1 and ref. [14]) are essential to detect and understand changes in this ecosystem and its functions.

Future climate change could cause the shift of ENSO and SAM toward higher frequencies of positive phases[9,41], which in turn might change the overall occupancy of certain feeding areas or prey resources by humpback whales on a hemisphere-wide spatial scale. Our results clearly show that acoustic detection of whales can shed light on biophysical interactions within the fascinating Southern Ocean ecosystem. Interannual trends in the distribution or health status (e.g., ref. [42]) of humpback whales and other baleen whales from the South Atlantic, but also other areas, warrant further investigation to provide information to whale stock and fishery management. Evaluating the sensitivity of keystone species to climate variabilities is essential to our understanding of the effects of climate-induced changes on the Southern Ocean ecosystem.

## Methods

**Passive acoustic data**. Humpback whale acoustic behavior throughout the ASSO was investigated by analyzing a multi-year passive acoustic dataset (2010–2018) from five recording positions along the Greenwich Meridian (Table 2, Fig. 1). Passive acoustic recordings were obtained using SonoVaults (Develogic GmbH, Hamburg) operated on a continuous recording scheme and with a sampling rate of 5,333 or 6,857 Hz (Table 2).

**Automatic detection and classification of humpback whale vocalizations**. All available passive acoustic data were processed by the "Low Frequency Detection and Classification System" (LFDCS) developed by ref. [43] and a custom-made acoustic-context filter to detect humpback whale acoustic presence at an hourly basis (humpback whales that did not produce any sounds remained undetected). LFDCS was set up with a customized call library based on the most common vocalization types of humpback whales and other acoustically abundant Antarctic

marine mammal species (i.e., Antarctic minke whale, killer whale (*Orcinus orca*), Weddell seal (*Leptonychotes weddellii*), crabeater seal (*Lobodon carcinophaga*), leopard seal (*Hydrurga leptonyx*), and Ross seal (*Ommatophoca rossii*)) [6,44–48]. Parameter settings and thresholds of LFDCS and the acoustic-context filter were tuned employing multiple test datasets to optimize the automatic detection of humpback whale vocalizations to the requirements of this study. Detailed information on set up and test runs of the automatic detection process is provided in the Supplementary material (Supplementary Note 1)[49].

**Manual post-processing of detection results**. In order to limit the temporal effort of manual post-processing, only even hours (i.e., hours starting at 00:00, 02:00, 04:00, 06:00, 08:00, 10:00, 12:00, 14:00, 16:00, 18:00, 20:00, and 22:00) were included in the further analysis. Four human analysts revised even hours with presumed humpback whale acoustic presence visually and aurally for the presence of humpback whale vocalizations by creating spectrograms in Raven Pro 1.5 (Hann Window, 1025–1790 window size, 80% overlap, 2048 DFT size[50]). Spectrograms were screened for humpback whale vocalizations by viewing windows of 60 s duration, spanning 0 to 1.80 kHz. Hours with confirmed humpback whale acoustic presence could contain both humpback whale social calls and humpback whale song.

**Environmental data**. The SIC data used for this study were extracted from a combination of satellite sensor data from the Nimbus-7 Scanning Multichannel Microwave Radiometer (SMMR), the Defense Meteorological Satellite Program (DMSP) -F8, -F11, and -F13 Special Sensor Microwave/Imrs (SSM/Is), and the DMSP-F17 Special Sensor Microwave Imager/Sounder (SSMIS), with a grid size of 25 km[51]. The data were used to calculate the daily SIC of the area within a 50 km radius around each recording location, with the Daily Antarctic Sea Ice Concentration packages in MATLAB[52]. The radius of 50 km was chosen because the acoustic range of humpback whales in the ASSO was estimated at 2–78 km[7].

The two most common climate indices for the Southern Hemisphere, the Southern Annular Mode (SAM)[53] and the Oceanic Niño Index (ONI, representing ENSO variabilities)[22] were used in this study. SAM data were downloaded from the Climate Data Guide[54] as monthly averages. ONI data were downloaded from the Climate Prediction Centre[22] as three-month running means.

### Table 2 Overview of passive acoustic data.

| Mooring ID | Latitude | Longitude | Recorder ID | Sampling frequency (Hz) | Deployment depth (m) | Recording period (total in days) |
|---|---|---|---|---|---|---|
| G1 (AWI227) | 59 2.82 °S | 000 5.78 °E | SV0002<br>SV1025<br>SV1004 | 5333<br>5333<br>6857 | 1007<br>1020<br>1070 | 2010-12-11–2011-05-21<br>2011-05-30<br>2011-06-14–2011-08-22<br>2012-12-11–2013-07-13<br>2016-12-22–2018-09-18<br>(1079) |
| G2 (AWI229) | 63 59.85 °S | 000 1.84 °E | SV1000<br>SV1010<br>SV1057 | 5333<br>5333<br>6857 | 1007<br>998<br>970 | 2010-12-15–2011-06-18<br>2012-12-14–2013-08-02<br>2014-12-16–2016-05-19<br>(936) |
| G3 (AWI230) | 66 2.01 °S | 000 3.12 °E | SV1001<br>SV1009 | 5333<br>5333 | 934<br>949 | 2010-12-16–2012-04-13<br>2012-05-06–2012-09-17<br>2013-01-07–2013-09-27<br>(881) |
| G4 (AWI231) | 66 30.71 °S | 000 1.51 °E | SV1002<br>SV1058<br>SV1023 | 5333<br>6857<br>6857 | 1083<br>973<br>859 | 2010-12-17–2012-02-05<br>2012-02-28–2012-07-30<br>2012-08-04–2012-08-09<br>2012-08-11–2012-08-14<br>2014-12-18–2016-05-28<br>2016-12-26–2018-10-28<br>(1776) |
| G5 (AWI232) | 68 59.94 °S | 000 4.38 °E | SV1003<br>SV1011<br>SV1059 | 5333<br>5333<br>6857 | 987<br>958<br>999 | 2010-12-18–2012-05-09<br>2012-06-01–2012-08-10<br>2012-12-17–2013-05-28<br>2013-06-19–2013-11-13<br>2015-01-08–2015-01-26<br>2015-02-14–2015-02-21<br>2015-03-04–2015-08-24<br>(1086) |

Information on passive acoustic recordings included in this study. The different recording periods at the five mooring positions were covered by different SonoVault recording units. For reference to earlier publications, the original mooring ID is listed in brackets.

**Statistics and reproducibility**. To assess the impact of the three climate variables, SIC, ONI, and SAM on the acoustic presence of humpback whales in the ASSO, generalized additive mixed models (GAMMs) were applied in R [49,55]. To model the effects of the three climate variables on the presence of humpback whales in the ASSO, the data from G3 and G4 were combined into a single time series (i.e., averaged daily SIC and daily averaged proportion of hours with humpback whale acoustic presence) because these recording positions were less than 50 km apart and provided the most complete time series (see Table 2 and Supplementary Note 2). SAM and ONI were also converted into categorical variables with negative, neutral, and positive phases to investigate responses both to small-scale and large-scale changes of climate variables (index value $< -0.5$ = negative; index value between $-0.5$ and $0.5$ = neutral; index value $> 0.5$ = positive; see [56] for details on ONI categories; the same standard was applied for SAM to create a neutral buffer between positive and negative phases). Binomial GAMMs were applied to model the daily acoustic presence/absence of humpback whales at G3/4 as a function of month, SIC, ONI (either continuous or categorical variable), and SAM (either continuous or categorical variable), including a model to account for temporal autocorrelation (functions *gamm* of the package *mgcv*[57] and *corARMA* of the package *nlme*[58] for an auto-regressive moving average (ARMA) model for the residuals). The optimal setup of starting values and orders for the implemented correlation structure was estimated in two ways: (1) with the function *auto.arima* (package *forecast*[59]), (2) by allowing the corARMA function to estimate its parameters directly from our data (used in the final model). The variables month and SIC were modeled with a cyclic smoothing term to account for the natural seasonal fluctuations. Model selection was performed using the Akaike Information Criterion (AIC), adjusted r-squared values, and the analyses of residuals.

**Reporting summary**. Further information on research design is available in the Nature Research Reporting Summary linked to this article.

## Data availability

The hourly humpback whale acoustic presence data that support the findings of this study are available on Dryad (https://doi.org/10.5061/dryad.ncjsxkss0) with the identifier https://doi.org/10.5061/dryad.ncjsxkss0[60].

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

## Acknowledgements

Thanks to Develogic GmbH, Hamburg, to the logistics department of the Alfred Wegener Institute, Bremerhaven, the mooring team of the AWI's physical oceanography department, to Reederei F. Laeisz GmbH, Rostock, and the crews of RV Polarstern expeditions ANT-XXVII/2, ANT-XXIX/2, PS89, PS103 for their contribution to the development, setup or maintenance of the passive acoustic recording array. We thank Maria Mallet and Katharina Hiemer for assistance with the manual post-processing of acoustic data and the whole team of the Ocean Acoustics Lab for the productive discussions on this study. We also want to thank Mark Baumgartner and Genevieve Davis for the assistance in setting up LFDCS and Bettina Meyer for discussions on Antarctic krill ecology.

## Author contributions

E.S. analyzed the data and wrote the manuscript. K.T. participated in some data collection and helped draft the manuscript. O.B. coordinated the study and collected the majority of the data. G.B. guided the analysis and helped draft the manuscript. S.M.W. helped with data analysis. A.E.-G. supervised the statistical analysis. I.V.O. coordinated the study, collected part of the data, and helped draft the manuscript. All the authors reviewed and contributed to the final document edits. All the authors gave the final approval for publication.

## Funding

## Competing interests

The authors declare no competing interests.
