## [Peer Review File · Communications Biology]

Reviewers' Comments:

Reviewer #1:

Remarks to the Author:

This is a fascinating manuscript. The data presented here are incredible: a series of good field indicators of humpback whales' occurrence over several years in an extremely remote part of the world. The authors then use an overlay of occurrence data with two climatic measures to develop an interesting argument that El Niño leads to humpbacks staying away from the sector of the Antarctic Atlantic that the authors have studied. They then relate this argument back to the relevant literature on humpback whales' biology, and to Antarctic oceanography. This work is exciting and interesting, and will be of great interest to anyone in marine ecology and conservation biology, especially given the growing awareness of whales' role as ecosystem engineers in the Antarctic system.

In the supplementary material, the authors also do a great job of demonstrating that their way of assessing humpback occurrence from their acoustic data is reliable, testing and truthing Baumgartner's LFDCS.

There's really only one substantive problem with the manuscript as it's written, and that is that there is no analytical model developed to assess the relationship between the climatic variables and humpbacks' detected occurrence. Without this analysis, and one that can account for the hiatus in the acoustic data in 2014/15 and to a lesser extent in 2016/17, the manuscript lacks mathematical demonstration that the signal that the authors have detected is real. Some form of time-series analysis, perhaps even just a GLM or GAM, is needed for the authors to make their point effectively. Without this, the manuscript lacks a solid foundation.

There are some much more minor issues as well:

1. In Figure 1, the W6-13 moorings should be removed. They're not relevant to the manuscript and need to go.
2. The field data available from mooring G5 (the coastal mooring) aren't enough to make any inferences. This is made clear in Figure 2 panel B. Apart from from some time in 2105, there were no recordings post-2013.
3. In Figure 2 panel C I can't see where it's stated which green line is the SAM and which is the ONI. Clarify please.
4. Lines 103-106: A more appropriate reference 17 would be Brown et al 1995, both for precedence and because it deals with Southern Hemisphere humpback whales. [Brown et al 1995 Evidence for a sex-segregated migration in the humpback whale (*Megaptera novaeangliae*) Proc. R. Soc. Lond. B.259229–234 <http://doi.org/10.1098/rspb.1995.0034>]

Reviewer #2:

Remarks to the Author:

This paper presents a time-series of humpback whale acoustic presence obtained from five mooring stations located at the eastern margin of the Weddell Sea, Antarctica. The major claim of the paper is that humpback whale presence is not driven by local sea ice conditions, but large-scale climate variability; whales are notably absent during the 2015-2016 period when two major climate oscillations (ENSO and SAM) are both in their positive phase.

While I enjoyed reading this paper, and to my knowledge the claims are novel, I am not convinced

the authors' argument is compelling. Indeed, I was surprised to find no numerical or statistical modelling in the paper to support the above claim. The paper presents biological and environmental time-series and proposes a hypothesis, but this is not supported by any quantitative analysis. The nature of the data does present some issues, but I provide some suggestions for analytical approaches below.

In my opinion, the paper would benefit from revision in a number of aspects:

1. The title is rather misleading and at risk of overselling the claims. "all year" is clearly not supported by the data, which show a very strong seasonality in whale acoustic presence (entirely consistent with current knowledge and expectations). "drives them out" implies some causative mechanism (e.g. advancement of dense ice or otherwise disadvantageous conditions) that is not actually articulated within this paper.

2. Relatedly, L74-75 "The major interannual variabilities seem to be driven by large-scale climate variabilities, but not local sea ice conditions". This statement remains an hypothesis, and is not substantiated in this paper. To provide strong evidence requires a numerical analysis. I understand that the 'acoustic presence' data presents difficulties, but one option to try and quantify the relationship with these drivers would be to identify presence/absence each day (or week) of year where data was available, and then fit a binomial model something like

presence \sim SIC + SAM + ONI

This could be for G1 (the offshore station with most data) or using all stations, including station as a random effect (eg. using the GAMM functionality within R package mgcv).

3. Similarly, L75-77 "local sea ice concentration is not the main environmental factor explaining the distribution...". One of the clearest signals evident within the data series is the annual seasonality. Again, rather than qualitatively dismissing sea ice, evaluating this relationship would be very useful; departure from the area annually does actually look very synchronous with the SIC. One option could be to determine 'departure day of year' as when acoustic detection falls below a certain level (e.g. proportion of 0.1 or 0.2 at a given station). Then, model this directly as a function of the SIC on that DOY, again considering appropriate random effects as above (noting that DOY 0 ought to be set to logically pre-date first detection).

4. Some tempering is required of the language around expectations regarding summer/wintertime presence, as well as improved referencing on this (L29-31, L77-78, L102 etc). The authors have previously published on winter persistence (10.1371/journal.pone.0073007). Satellite tracking data has also previously demonstrated lengthy residence in the Peninsula regions nearby (eg. Weinstein et al 2017, doi: 10.1016/j.biocon.2017.04.014, documented commencement of migration northward >60S ranging from March 22nd to July 17th, mean = May 25th) and elsewhere (e.g. Andrews-Goff et al. 2018, doi: 10.1038/s41598-018-30748-4, tags transmitted in Antarctic waters as late as May 31st prior to tag failure). Out of the five years of acoustic data, there is no call data past June in 2012 and 2013; a tiny amount in July 2017 and 2018 at the farthest offshore station G1(actual number of calls not given?); but more substantive presence occurs during July and even August in 2011 (again at offshore G1). Are these large number of late calls associated with low ice at G1 in 2011 (as per pt 3 above)?

Overall, the acoustic data appear consistent with expectations that presence begins to decline as ice advances annually. Whales mainly disappear (cease recording calls) by 80% SIC except for 2011? From an interannual perspective, the anomalous 2015-2016 seasons appear to represent an almost complete absence i.e. non-arrival to the area, potentially determined by other factors (to be tested as above); but this ought not be framed as being driven out (as they were never in...)

If the claims can be further substantiated, the paper has potential to influence thinking in the field, since there is widespread interest in linking marine megafauna migrations with climate signatures.

For the record I am not an acoustician. To my reading the methodology of the acoustic processing appears to be clearly documented but I leave other reviewers with this expertise to critically review this aspect of the paper.

General comments

Acoustic presence supplies a record of calling whales (i.e. there may be non-calling whales present). Somewhere in the paper this needs to be clearly articulated, and some statement made regarding whether this species is expected/known to call year-round, change calls between breeding/feeding seasons, etc. Also, for readers interpretation of Figure 2, that a very high acoustic presence (e.g. May 2011) may be representing many animals OR the same animals calling many times and indeed moving between stations.

The authors are to be commended for maintaining the field array over multiple seasons – this is no small feat. These data are valuable for the research field, and the paper outcomes should be of interest to others in the field. The lack of long-term monitoring is a fundamental and widespread issue with most ecological systems, especially the remote Antarctic where logistical costs are high. While the time-series obtained is commendable, it is still too short to capture more than one oscillation in the major climate signal. This should be clearly noted within the manuscript and used to support the call for ongoing maintenance of such observational arrays. While still requiring ship deployment and retrieval, such automated stations represent relatively cost-efficient ecological monitoring.

The paper is very well written and succinct. There are a few minor discrepancies in the English grammar, particularly the last two paragraphs of the main manuscript, but I leave that to be covered by the editorial process.

Specific comments

L20. Again, wording misleads reader to expect year-round presence.

L43-48 is a very lengthy sentence that needs to be broken apart.

L65. Just to set the readers at ease, it would be worthwhile to supply some data that convincingly demonstrates the mooring stations were operational within the important 2015-16 seasons. This might for example be a supplementary table that documents the total number of calls of all species per season for each year (2011-2018), thereby showing other species (but not humpbacks) continued to be recorded during this period.

L84-86. Needs reorganising, as the references given are mainly for the physical processes listed, but the sentence reads as if they all support knowledge of physical influences upon krill.

L87. "Outside" – north or south?

L106-110. Needs some clarification as to which HBW populations (stocks) are expected to be feeding in the study area.

L115. "Food web interactions of" is not really covered here; how about "biophysical interactions within"?

Final paragraph starting L112. The use of animal health / body condition indices is also very promising for investigating interannual trends and consequences eg. Bengston-Nash et al. 2018 doi: 10.1111/gcb.14035.

Methods L163. It would be useful to present some ice maps, perhaps in the Supplement?

Figure 1. It would be useful to supply contours for the minimum and maximum sea ice extent recorded during the study period on this map. Since all the latest records within a season are usually from the outermost station G1, it would be helpful to know if this lies within/outside the sea ice.

Figure 2. The spacing of the dashed vertical lines is unclear. It would be very helpful if these

clearly demarcated seasons/quarters (ie Jan 1, Apr 1, Jul 1, Oct 1) so the months are easy to identify. On panel 2A, it is also confusing to have a y-axis with 'Proportion' but maximum value at 2. In the caption it needs to clarify that complete presence at all four stations would give a value of $y=4$.

Throughout paper: since 'presence' is not equivalent to 'distribution' (studies via sightings surveys, tracking etc) suggest careful choices with this wording.

Response reviewer comments

REVIEWER COMMENTS

#1

There's really only one substantive problem with the manuscript as it's written, and that is that there is no analytical model developed to assess the relationship between the climatic variables and humpbacks' detected occurrence. Without this analysis, and one that can account for the hiatus in the acoustic data in 2014/15 and to a lesser extent in 2016/17, the manuscript lacks mathematical demonstration that the signal that the authors have detected is real. Some form of time-series analysis, perhaps even just a GLM or GAM, is needed for the authors to make their point effectively. Without this, the manuscript lacks a solid foundation.

Reply – Due to the request of both reviewers to implement a model that would support our hypothesis, we explored a variety of modelling options, including GLMMs and GAMMs, to test our hypothesis without violating any of the statistical assumptions. Finally, we implemented binomial GAMMs and included a function to account for the temporal autocorrelation of the data (corARMA). See line 313-343 for the detailed description of the method:

Statistics and Reproducibility

To assess the impact of the three climate variables, SIC, ONI, and SAM on the acoustic presence of humpback whales in the ASSO, generalized additive mixed models (GAMMs) were applied in R⁵⁵. To model the effects of the three climate variables on the presence of humpback whales in the ASSO, the data from G3 and G4 were combined into a single time series (i.e., averaged SIC and averaged proportion of hours with humpback whale acoustic presence) because these recording positions were less than 50km apart and provided the most complete time series. SAM and ONI were also converted into categorical variables with negative, neutral and positive phases (index value < -0.5 = negative; index value between -0.5 and 0.5 = neutral; index value > 0.5 = positive; see Climate Prediction Centre³¹ for details on ONI categories, the same standard was applied for SAM to create a neutral buffer between positive and negative phases). Binomial GAMMs were applied to model the daily acoustic presence/absence of humpback whales at G3/4 as a function of Month, SIC, ONI (either continuous or categorical variable), and SAM (either continuous or categorical variable), including a model to account for temporal autocorrelation (function `gamm` of the package `mgcv`⁵⁶ and function `corARMA` of the package `nlme`⁵⁷). The function `auto.arima` (package `forecast`⁵⁸) was used to find the optimal setup of starting values and orders for the implemented correlation structure. Model selection was performed using the Akaike Information Criterion (AIC), adjusted R² values, and the analyses of residuals.

and line 14-158, Table 1, and Figure 4 for the description of the results:

Modeling the effect of sea ice concentration, SAM, and ONI, on the acoustic presence of humpback whales at the study location revealed that all three environmental variables explain the observed patterns in the humpback whale acoustic presence in the ASSO. The best-fit model explaining most of the variability in the data included month, sea ice concentration, SAM as continuous predictors and ONI as a categorical predictor ($R^2 = 0.52$; Table 1). The smoothed effects of month and sea ice concentration were highly significant, because these variables explain seasonality in humpback whale presence on the feeding ground (Table 1, Figure 3; ^{7,12}) in five out of seven years. The model shows that the ONI in the positive phase predicts a significantly lower probability of humpback whale acoustic presence than ONI in a neutral or negative phase (Table 1, Figure 3). The smoothed effect of the SAM index was also statistically significant (Table 1, Figure 3). The model prediction for the SAM index shows lower predicted values at negative and high positive index values, although with higher uncertainties (Figure 3). This appears reasonable when looking at the original time-series. The negative phases of SAM were usually registered during winter when acoustic presences are naturally low, and extreme positive phases were only registered during summer 2015 and 2016 (Figure 2). Uncertainties at extreme index values (also for ONI) are high because these values are rare in the analyzed time-series and a much longer time-series would probably be required to quantify the relationship between humpback whale presence and climate indices with higher certainty.

Table 1. Summary of the best-fit model, explaining most of the data variability, including sea ice concentrations (SIC), SAM, and month as smooth terms, as well as ONI as a categorical predictor. Note that the factor levels of ONI as a categorical predictor are listed under the parametric coefficients and that the 'ONICPositive' level is represented by the intercept.

Formula:				
PA ~ s(SIC) + ONIc + s(SAM) + s(Month)				
Parametric coefficients:				
	Estimate	Std. Error	t value	Pr(> t)
(Intercept)	-7.1071	0.8424	-8.436	< 2e-16 ***
ONICNeutral	4.6147	0.8212	5.620	2.17e-08 ***
ONICNegative	5.4203	0.8863	6.116	1.15e-09 ***
Approximate significance of smooth terms:				
	edf	Ref.df	F	p-value

s(SIC)	4.037	4.037	15.791	8.48e-13 ***
s(SAM)	5.602	5.602	2.941	0.00726 **
s(Month)	5.523	8.000	8.656	1.88e-15 ***
R-sq.(adj) = 0.519				

Figure 3. Model predictions of the best-fit model, including the smooth terms sea ice concentration (SIC), month, and SAM as well as ONI as a categorical predictor (see methods for further explanation of categories). Grey shaded areas depict 95%-confidence intervals.

There are some much more minor issues as well:

1. In Figure 1, the W6-13 moorings should be removed. They're not relevant to the manuscript and need to go.

Reply – We understand the point of the reviewer, but we would like to keep the additional recording positions in the plot as they are relevant to our plea for continued and expanded long-term monitoring to understand these long-term patterns in behavior. Furthermore, we believe that the coloring and caption provides sufficient clarity to the reader that the grey sites are not included in the analyses, but form part of the greater network with which the data were collected. We refer to those in the discussion in line 206 to point out the importance of maintaining long-term observation systems.

2. The field data available from mooring G5 (the coastal mooring) aren't enough to make any inferences. This is made clear in Figure 2 panel B. Apart from from some time in 2105, there were no recordings post-2013.

Reply – We agree with the reviewer that the data from this position are not that abundant. However, the given fact of absence of calls recorded during 2015 in the G5 data, whereas there were (albeit sparse) calls during the same period in previous years at G5, to us supports the overall findings and our decision to keep G5 included. Based on the minor role that these data play in the overall analyses, we have decided to put this additional figure in the appendix.

3. In Figure 2 panel C I can't see where it's stated which green line is the SAM and which is the ONI. Clarify please.

Reply – In the new Figure 2, ONI and SAM are represented as orange and green lines and the legend was placed directly on-top of the panel B (previous panel C).

Figure 2. A: Average proportion of hours with humpback whale acoustic presence per month from the four oceanic recording positions (G1-G4) on the Greenwich Meridian from December 2010 until September 2018. Grey bars represent months without recording data. The blue solid line and the right y-axis depict the daily averaged sea ice concentration within a 50km radius around each recording location. **B:** Climatic variations from 2011 until 2018 indicated by three-month running means of the Southern Annular Mode index (SAM) as a dominant pattern of natural climate variability in polar and subpolar regions of the Southern Hemisphere and the Oceanic Niño Index (ONI) representing the periodic fluctuation of sea surface temperature and air pressure originating from the tropical Pacific. Time span of strong El Niño phase in 2015/16 is indicated by the yellow rectangle.

4. Lines 103-106: A more appropriate reference 17 would be Brown et al 1995, both for precedence and because it deals with Southern Hemisphere humpback whales. [Brown et al 1995 Evidence for a sex-segregated migration in the humpback whale (Megaptera novaeangliae)Proc. R. Soc. Lond. B.259229–234 <http://doi.org/10.1098/rspb.1995.0034>]

Reply – We additionally included the recommended reference in line 214 as the references show complementary evidences, with Brown et al. 1995 presenting information from the

Southern Hemisphere and Craig et al. 2003 presenting temporal estimates of timing of migration.

#2

While I enjoyed reading this paper, and to my knowledge the claims are novel, I am not convinced the authors' argument is compelling. Indeed, I was surprised to find no numerical or statistical modelling in the paper to support the above claim. The paper presents biological and environmental time-series and proposes a hypothesis, but this is not supported by any quantitative analysis. The nature of the data does present some issues, but I provide some suggestions for analytical approaches below.

Reply – The main problems of our data for statistical analysis are the existence of data gaps and the temporal autocorrelation among hourly or daily samples of acoustic presence and environmental conditions (here, sea ice concentration). Additionally, our time-series data only covers one El Niño event which probably would not be sufficient to support a statistical model. However, responding to the request of both reviewers, we now include a model for our data in the revised version of the manuscript (see below, reply to comment 2).

In my opinion, the paper would benefit from revision in a number of aspects:

1. The title is rather misleading and at risk of overselling the claims. “all year” is clearly not supported by the data, which show a very strong seasonality in whale acoustic presence (entirely consistent with current knowledge and expectations). “drives them out” implies some causative mechanism (e.g. advancement of dense ice or otherwise disadvantageous conditions) that is not actually articulated within this paper.

Reply – We understand the point of the reviewer and changed the title to a shorter and hopefully less misleading version (line 1-2).

Multi-year presence of humpback whales in the Atlantic sector of the Southern Ocean but not during El Niño

2. Relatedly, L74-75 “The major interannual variabilities seem to be driven by large-scale climate variabilities, but not local sea ice conditions”. This statement remains an hypothesis, and is not substantiated in this paper. To provide strong evidence requires a numerical analysis. I understand that the ‘acoustic presence’ data presents difficulties, but one option to try and quantify the relationship with these drivers would be to identify presence/absence each day (or week) of year where data was available, and then fit a binomial model something like

presence ~ SIC + SAM + ONI

This could be for G1 (the offshore station with most data) or using all stations, including station as a random effect (eg. using the GAMM functionality within R package mgcv).

Reply – We implemented GAMMs in order to quantify the relationship between climate variables and humpback whale acoustic presence (line 314-339). To deal with the variability in temporal coverage among the different recording locations (see Supplementary Material S2), we merged the acoustic presence data of the recording location G3 and G4 into a single time series (see line 316). This was possible because these recording positions were less than 50km apart, which is within the range of acoustic detection for humpback whales in the area (see Van Opzeeland et al. 2013), and sea ice concentrations were almost identical between both sites (less than 10% divergence).

Statistics and Reproducibility

To assess the impact of the three climate variables, SIC, ONI, and SAM on the acoustic presence of humpback whales in the ASSO, generalized additive mixed models (GAMMs) were applied in R⁵⁵. To model the effects of the three climate variables on the presence of humpback whales in the ASSO, the data from G3 and G4 were combined into a single time series (i.e., averaged SIC and averaged proportion of hours with humpback whale acoustic presence) because these recording positions were less than 50km apart and provided the most complete time series. SAM and ONI were also converted into categorical variables with negative, neutral and positive phases (index value < -0.5 = negative; index value between -0.5 and 0.5 = neutral; index value > 0.5 = positive; see Climate Prediction Centre³¹ for details on ONI categories, the same standard was applied for SAM to create a neutral buffer between positive and negative phases). Binomial GAMMs were applied to model the daily acoustic presence/absence of humpback whales at G3/4 as a function of Month, SIC, ONI (either continuous or categorical variable), and SAM (either continuous or categorical variable), including a model to account for temporal autocorrelation (function gamm of the package mgcv⁵⁶ and function corARMA of the package nlme⁵⁷). The function auto.arima (package forecast⁵⁸) was used to find the optimal setup of starting values and orders for the implemented correlation structure. Model selection was performed using the Akaike Information Criterion (AIC), adjusted R² values, and the analyses of residuals.

The results of the best-fit model explaining most of the variability are presented in Table 1, Figure 4 and lines 140-158:

Modeling the effect of sea ice concentration, SAM, and ONI, on the acoustic presence of humpback whales at the study location revealed that all three environmental variables explain the observed patterns in the humpback whale acoustic presence in the ASSO. The best-fit model explaining most of the variability in the data included month, sea ice concentration, SAM as continuous predictors and ONI as a categorical predictor ($R^2 = 0.52$;

Table 1). The smoothed effects of month and sea ice concentration were highly significant, because these variables explain seasonality in humpback whale presence on the feeding ground (Table 1, Figure 3; ^{7,12}) in five out of seven years. The model shows that the ONI in the positive phase predicts a significantly lower probability of humpback whale acoustic presence than ONI in a neutral or negative phase (Table 1, Figure 3). The smoothed effect of the SAM index was also statistically significant (Table 1, Figure 3). The model prediction for the SAM index shows lower predicted values at negative and high positive index values, although with higher uncertainties (Figure 3). This appears reasonable when looking at the original time-series. The negative phases of SAM were usually registered during winter when acoustic presences are naturally low, and extreme positive phases were only registered during summer 2015 and 2016 (Figure 2). Uncertainties at extreme index values (also for ONI) are high because these values are rare in the analyzed time-series and a much longer time-series would probably be required to quantify the relationship between humpback whale presence and climate indices with higher certainty.

Table 2. Summary of the best-fit model, explaining most of the data variability, including sea ice concentrations (SIC), SAM, and month as smooth terms, as well as ONI as a categorical predictor. Note that the factor levels of ONI as a categorical predictor are listed under the parametric coefficients and that the 'ONICPositive' level is represented by the intercept.

Formula:				
PA ~ s(SIC) + ONIC + s(SAM) + s(Month)				
Parametric coefficients:				
	Estimate	Std. Error	t value	Pr(> t)
(Intercept)	-7.1071	0.8424	-8.436	< 2e-16 ***
ONICNeutral	4.6147	0.8212	5.620	2.17e-08 ***
ONICNegative	5.4203	0.8863	6.116	1.15e-09 ***
Approximate significance of smooth terms:				
	edf	Ref.df	F	p-value
s(SIC)	4.037	4.037	15.791	8.48e-13 ***
s(SAM)	5.602	5.602	2.941	0.00726 **
s(Month)	5.523	8.000	8.656	1.88e-15 ***
R-sq.(adj) = 0.519				

Figure 3. Model predictions of the best-fit model, including the smooth terms sea ice concentration (SIC), month, and SAM as well as ONI as a categorical predictor (see methods for further explanation of categories). Grey shaded areas depict 95%-confidence intervals.

3. Similarly, L75-77 “local sea ice concentration is not the main environmental factor explaining the distribution...”. One of the clearest signals evident within the data series is the annual seasonality. Again, rather than qualitatively dismissing sea ice, evaluating this relationship would be very useful; departure from the area annually does actually look very synchronous with the SIC. One option could be to determine ‘departure day of year’ as when acoustic detection falls below a certain level (e.g. proportion of 0.1 or 0.2 at a given station). Then, model this directly as a function of the SIC on that DOY, again considering appropriate random effects as above (noting that DOY 0 ought to be set to logically pre-date first detection).

Reply – Previous studies (i.e., references line 147) and the applied model clearly show that sea ice concentration is an important factor explaining the seasonality of humpback whale presence on the feeding grounds. However, this is nothing new and is not the main focus of this paper. Therefore, we decided to only include the results of the binomial GAMMs explained above, which also clearly present a significant effect of sea ice concentration due to its importance in explaining seasonality in five out of seven years of sampling effort (years with more frequent acoustic detections; neutral and negative ONI phases). We added a sentence and additional references in line 144-147 to clarify this to the reader.

The smoothed effects of month and sea ice concentration were highly significant, because these variables explain seasonality in humpback whale presence on the feeding ground (Table 1, Figure 3; ^{7,12}) in five out of seven years.

4. Some tempering is required of the language around expectations regarding summer/wintertime presence, as well as improved referencing on this (L29-31, L77-78, L102 etc). The authors have previously published on winter persistence

(10.1371/journal.pone.0073007). Satellite tracking data has also previously demonstrated lengthy residence in the Peninsula regions nearby (eg. Weinstein et al 2017, doi: 10.1016/j.biocon.2017.04.014, documented commencement of migration northward >60S ranging from March 22nd to July 17th, mean = May 25th) and elsewhere (e.g. Andrews-Goff et al. 2018, doi: 10.1038/s41598-018-30748-4, tags transmitted in Antarctic waters as late as May 31st prior to tag failure). Out of the five years of acoustic data, there is no call data past June in 2012 and 2013; a tiny amount in July 2017 and 2018 at the farthest offshore station G1(actual number of calls not given?); but more substantive presence occurs during July and even August in 2011 (again at offshore G1).

Are these large number of late calls associated with low ice at G1 in 2011 (as per pt 3 above)?

Reply – We thank the reviewer for pointing this out. Therefore, we have included the new Figure 2 and (Supplementary Material) S2 that have an improved visualization of winter presence, referring to acoustic presence during September, October, and November (see S2, G3 and G5 in 2011). Thus, the recommended references would not sufficiently support our argument and were therefore not included.

The recording position G1 is very close to the sea ice edge during winter, which can be seen in the ice maps from the appendix (Supplementary Material S3) which were supplied upon the recommendation of the reviewer. This vicinity to the sea ice edge probably explains in large parts the acoustic presence of whales until later in the year at this position, but also its geographic position along the migration route could be an alternative explanation.

Figure 2. **A:** Average proportion of hours with humpback whale acoustic presence per month from the four oceanic recording positions (G1-G4) on the Greenwich Meridian from December 2010 until September 2018. Grey bars represent months without recording data. The blue solid line and the right y-axis depict the daily averaged sea ice concentration within a 50km radius around each recording location. **B:** Climatic variations from 2011 until 2018 indicated by three-month running means of the Southern Annular Mode index (SAM) as a dominant pattern of natural climate variability in polar and subpolar regions of the Southern Hemisphere and the Oceanic Niño Index (ONI) representing the periodic fluctuation of sea surface temperature and air pressure originating from the tropical Pacific. Time span of strong El Niño phase in 2015/16 is indicated by the yellow rectangle.

Overall, the acoustic data appear consistent with expectations that presence begins to decline as ice advances annually. Whales mainly disappear (cease recording calls) by 80% SIC except for 2011? From an interannual perspective, the anomalous 2015-2016 seasons appear to represent an almost complete absence i.e. non-arrival to the area, potentially determined by other factors (to be tested as above); but this ought not be framed as being driven out (as they were never in...)

Reply – We actually cannot tell from our data if the whales never arrived or arrived and left again, so we changed the wording ‘driven out’ in the title. As the reviewer states, the acoustic data is consistent with the expectation that presence begins to decline when sea ice concentration increases and that whales mainly disappear at sea ice concentrations above 80%. Except for a few exceptions in the data (e.g., winter presences), this is also true for 2011, because sea ice concentrations in July and August 2011 were only a few percent higher than 80%, which is still in the range of the error for the calculation of sea ice concentration from satellite data (see, for example, [Spren, G., Kaleschke, L., & Heygster, G. (2008). Sea ice remote sensing using AMSR-E 89 GHz channels. *Journal of Geophysical Research: Oceans*, 113(C2)] for details).

If the claims can be further substantiated, the paper has potential to influence thinking in the

field, since there is widespread interest in linking marine megafauna migrations with climate signatures.

For the record I am not an acoustician. To my reading the methodology of the acoustic processing appears to be clearly documented but I leave other reviewers with this expertise to critically review this aspect of the paper.

General comments

Acoustic presence supplies a record of calling whales (i.e. there may be non-calling whales present). Somewhere in the paper this needs to be clearly articulated, and some statement made regarding whether this species is expected/known to call year-round, change calls between breeding/feeding seasons, etc. Also, for readers interpretation of Figure 2, that a very high acoustic presence (e.g. May 2011) may be representing many animals OR the same animals calling many times and indeed moving between stations.

Reply – We added additional information about this matter in the introduction, the results and the methods. In the introduction we added a sentence clarifying the extents of vocal activity in humpback whales as (line 43-44):

Humpback whales are excellent candidates for PAM studies due to their year-round vocal activity of all sex and age classes⁵⁻⁷.

In the results section we added information explaining high acoustic presences in Figure 3 and how this relates to the amount of whales present (line 96-101):

The high proportions of hours with humpback whale acoustic presence during autumn at G1-G4 coincided with the commencement of singing behavior in Southern Hemisphere humpback whale males^{18,19}. During this period, two or more individuals often were vocalizing at the same time and acoustic activity was registered at all four locations in parallel, indicating the presence of multiple animals.

In the methods section we added a statement in line 274-275 to clarify that only vocalizing humpback whales could be detected with our method:

(humpback whales which did not produce any sounds remained undetected)

The authors are to be commended for maintaining the field array over multiple seasons – this is no small feat. These data are valuable for the research field, and the paper outcomes should be of interest to others in the field. The lack of long-term monitoring is a fundamental and widespread issue with most ecological systems, especially the remote Antarctic where logistical costs are high. While the time-series obtained is commendable, it is still too short to capture more than one oscillation in the major climate signal. This should be clearly noted within the manuscript and used to support the call for ongoing maintenance of such observational arrays. While still requiring ship deployment and retrieval, such automated stations represent relatively cost-efficient ecological monitoring.

Reply – We thank the reviewer for this comment and added as recommended two sentences in line 238-241 that clearly communicate this limitation of our data to the reader:

The maintenance and implementation of further long-term observation systems such as the HAFOS mooring network from which the analyzed recordings originate (see Figure 1 and ¹⁴) are essential to detect and understand changes in this ecosystem and its functions.

The paper is very well written and succinct. There are a few minor discrepancies in the English grammar, particularly the last two paragraphs of the main manuscript, but I leave that to be covered by the editorial process.

Specific comments

L20. Again, wording misleads reader to expect year-round presence.

Reply – As stated above, there was actually year-round presence at G3 and G5 during some years. We adapted Figure 2 and added a Figure in the appendix (Supplementary Material S2) to provide a better visualization of this.

L43-48 is a very lengthy sentence that needs to be broken apart.

Reply – We shortened the respective sentence and rearranged the information in this part of the introduction, see line 44-56:

To improve the understanding of the ecological conditions under which humpback whales use the area as a feeding ground, we investigated the inter-annual changes in humpback whale acoustic presence in relation to three environmental parameters key to the Southern Ocean. The Southern Annular Mode (SAM) which is the dominant pattern of natural climate variability in polar and subpolar regions of the Southern Hemisphere, as well as the El Niño Southern Oscillation (ENSO) causing periodic fluctuation of sea surface temperature and air pressure originating from the tropical Pacific are climate oscillations which have large effects on the Southern Ocean productivity⁸⁻¹¹. Local sea ice concentration has direct effects on the whales' access to open water areas necessary for breathing and indirect effects through its impact on productivity^{7,8,12,13}. The large-scale mooring network in the ASSO that we have been maintaining for more than ten years¹⁴ allowed us to relate the long-term trend in humpback whale acoustic presence to long-term trends in sea ice concentration and climate oscillations.

L65. Just to set the readers at ease, it would be worthwhile to supply some data that convincingly demonstrates the mooring stations were operational within the important 2015-16 seasons. This might for example be a supplementary table that documents the total number of calls of all species per season for each year (2011-2018), thereby showing other species (but not humpbacks) continued to be recorded during this period.

Reply – We included some information and an additional reference in line 119-121. The analysis from this publication details the detections of Antarctic Minke whale vocalizations

also during the years 2015/16. This reference therewith demonstrates that the recording devices were operational during these years.

[...], whereas vocalization of other species, e.g., the Antarctic Minke whale, were detected during this time (see²⁰).

L84-86. Needs reorganising, as the references given are mainly for the physical processes listed, but the sentence reads as if they all support knowledge of physical influences upon krill.

Reply – We rearranged references in this sentence (see line 181) and also added more information and references in the following lines (191-204).

Epecially north of the Antarctic Polar Front, sea surface temperature increases and chlorophyll concentration decreases during positive SAM^{30,31}. ENSO has the strongest effects on the Pacific sector of the Southern Ocean, including the Western Antarctic Peninsula. The Western Antarctic Peninsula is a key habitat for Antarctic krill as a spawning and nursery ground, from which krill is transported with the Antarctic Circumpolar Current (ACC) ‘conveyor belt’ into north-eastern areas of the ASSO⁸. During or after the positive phase of ENSO, El Niño caused warming and the reduction of sea ice at the Western Antarctic Peninsula. Due to these climate conditions, less krill might be recruited from the Antarctic Peninsula towards the oceanic regions of the ASSO, although this process is probably delayed by multiple months considering the estimated transport speed of the ACC^{8,10,11,32}. Also, during years of El Niño, a manifestation of negative sea surface temperature anomalies in the southwest Atlantic, termed the Antarctic dipole, is common which probably affects productivity in this area^{10,11,30}.

L87. “Outside” – north or south?

Reply – „outside“ was changed to „north of” in line 182.

L106-110. Needs some clarification as to which HBW populations (stocks) are expected to be feeding in the study area.

Reply – We added some information and a reference in brackets in line 215-216:

(probably individuals from breeding stocks from South America and Africa³⁷)

L115. “Food web interactions of” is not really covered here; how about “biophysical interactions within”?

Reply – We thank the reviewer for the suggestion and we adjusted the text in line 245-246 accordingly.

Final paragraph starting L112. The use of animal health / body condition indices is also very promising for investigating interannual trends and consequences eg. Bengston-Nash et al. 2018 doi: 10.1111/gcb.14035.

Reply – We thank the reviewer for this suggestion and included the reference in line 213.

Methods L163. It would be useful to present some ice maps, perhaps in the Supplement?

Reply – We added additional supplementary material (S3) that shows monthly ice maps of the area from December 2010 to October 2018 including the monthly averaged acoustic presences at the recording position G1-G5.

Figure 1. It would be useful to supply contours for the minimum and maximum sea ice extent recorded during the study period on this map. Since all the latest records within a season are usually from the outermost station G1, it would be helpful to know if this lies within/outside the sea ice.

Reply – We understand the point of the reviewer but decided to keep the original map without the minimum and maximum ice extent as these values can vary strongly between years. Instead, we have provided monthly mean sea ice maps for all years in the electronic supplement (Supplementary Material S3) including the averaged acoustic presences at each recording position which also provides the requested information. The latest records of humpback whale acoustic presence within the year were at positions G3 and G5, so within the sea ice, this is now also visible from Figures 2 and 3 in the main text.

Figure 2. The spacing of the dashed vertical lines is unclear. It would be very helpful if these clearly demarcated seasons/quarters (ie Jan 1, Apr 1, Jul 1, Oct 1) so the months are easy to identify. On panel 2A, it is also confusing to have a y-axis with 'Proportion' but maximum value at 2. In the caption it needs to clarify that complete presence at all four stations would give a value of y=4.

Reply – The dashed vertical lines mark the start and end of seasons/quarters in the mid of December, March, June, and September in accordance with the timing of the longest and the shortest day of the year on December 21st and June 21st, respectively. Therefore, we consider the demarcation of seasons as correct. In the new Figure, we averaged the proportions over the four oceanic recording locations (range 0-1) to avoid confusion of the reader.

Throughout paper: since 'presence' is not equivalent to 'distribution' (studies via sightings surveys, tracking etc) suggest careful choices with this wording.

Reply – We checked our manuscript for discrepancies in this matter and believe that the results of large-scale (temporal and spatial) studies on presence can still be interpreted as results on distribution. Therefore, we changed the wording in our results section accordingly (see line 171) but continued using the term 'distribution' in the discussion to refer to potential ocean basin wide changes.

Reviewers' Comments:

Reviewer #2:

Remarks to the Author:

The authors have dealt appropriately with my concerns expressed on their previous draft of this manuscript. I have no further comments.

Reviewer #3:

Remarks to the Author:

In general, I find the authors have done a commendable job incorporating most of my points and those of the other reviewer. This includes moderating the title, adding a new statistical analysis, clarifying the nature of HBW calling behaviour, and adding supplementary figures. Nevertheless, I do have some issues with how the new analysis was conducted and am still concerned that the interpretations regarding the role of sea ice (as raised previously) have not been sufficiently moderated. These two concerns are elaborated below, but I do believe that with another round of revision these issues can be addressed, and a very interesting paper published.

1. Statistical analysis

I am not clear why the (most abundant?) acoustic data from G1 were excluded. But, given that the analysis was applied to blended data from G3/G4 I am not sure that a mixed-effect model is necessary. If station is not required as a random effect – what random effect is included? – it may be better to just simplify to a GAM.

This is very pertinent, because the text indicates that through use of the `mgcv` `gamm` function a temporal correlation term (`corARMA` from the `nlme` package interfaced with `mgcv`) has been included. However, the `mgcv` package includes a clear warning on this:

WARNINGS... gamm performs poorly with binary data, since it uses PQL. It is better to use gam with s(...,bs="re") terms, or gamm4. gamm assumes that you know what you are doing!

The package indicates that the preferred (most numerically robust) approach for binomial GAMMs is via package `gamm4` (also from Simon Wood) which interfaces to `lme4`; noting that `nlme` correlation structures are not able to be included for these. From my reading however, this current study could avoid the dubious call to `gammPQL` (which provides only approximate MLEs) by simplifying to a GAM (which will also enable the `corARMA` correlation).

Aside from this main point, it would be helpful to provide clarification about:

- why G1 was dropped from the analysis if this station seems to have the most abundant data? I would suggest repeating the same model (separately) for G1. If indeed this station shows no climate effects, this is very interesting and suggests whales still migrate past ASSO but do not enter.
- why it was necessary to explore categorical versions of the SAM and ONI covariates, rather than just use the continuous variables.
- is month necessary to include in the model if SIC essentially reflects season? Aren't these two covariates highly correlated?
- the SAM term is relatively weak (p-value c.f. other terms) and looks at risk of being overfitted. To defend its inclusion authors might consider using the dredge function from the package `MuMIn` to evaluate all term combinations and use model ranking.

2. Interpretation regarding the role and dynamics of ASSO sea ice.

There are still several statements in the manuscript that need to be moderated or deleted:

L107-109. "Sea ice concentration during 2015 and 2016 did not show strong anomalies in the area of the Greenwich Meridian (Figure 2A)."

L140-144. "The major inter-annual variabilities in the acoustic presence of humpback whales on a Southern Ocean feeding ground seem to be driven by large-scale climate variabilities, but not local sea ice conditions. Our new data indicate that the local sea ice concentration is not the main environmental factor explaining the presence of humpback whales in the Southern Ocean".

L146-148. "Furthermore, the years 2015 and 2016 did not show strong abnormalities in the local sea ice concentrations at the Greenwich Meridian, which suggests that sea ice concentration was not the driver for the humpback whale acoustic absence in these years".

As requested in the previous revision round, please reword this interpretation. It is not correct as currently phrased. There are no calculations in this manuscript pertaining to sea ice anomalies (at stations, or more broadly across the ASSO), so these statements are unsubstantiated (i.e. simply based on eyeballing Fig 2 and S3). In fact, the sea ice arrival looks very late in 2016. It is quite possible that the interannual climate oscillations (SAM/ONI) drive changes in ASSO ice extent, timing and duration; but this is not assessed here. In fact, authors state later in the paper: L156. "During positive values of SAM, the oceanic feeding areas of humpback whales in the ASSO show signs of reduced sea ice extent..." Exactly – these dynamics are probably linked – but you cannot say this in the Discussion after what you have said before!

I would suggest reformulating this interpretation along the lines of 1) SIC clearly plays a role in the intra-annual timing of whale presence (acoustic data is consistent with the expectation that presence begins to decline when sea ice concentration increases and that whales mainly disappear at sea ice concentrations above 80% (Figure 3)). 2) New finding: additionally, large-scale climate oscillations moderate the inter-annual presence of whales in the ASSO, going on to discuss what environmental changes these might drive (including ice dynamics). Statements about no abnormalities/anomalies need to be removed, as it is not assessed in the paper (unless the authors want to explore calculations of daily SIC anomalies relative to the climatological mean, at stations and/or across the ASSO).

Specific comments

Winter presence – this is still not evident up front in Fig 2A (to my eye) and while I appreciate the authors efforts to better include information on this in the new S2 and S3, unfortunately I still find myself squinting and wondering about better delivery of the data on year-round presence. A couple of suggestions might be compiling the 90+ maps in Supplementary S3 into a single animation file; developing a heatmap figure focusing on winter-spring months only (June-Nov) with station as column, month as row and heat colourmap indicating for example the number of days per month in which acoustic presence was detected; and/or a Table just documenting this information (for those years where Jun-Nov acoustic obs of HBW occurred). In any case these winter observations do seem to be extremely limited, indicating year-round presence to be the exception rather than the norm for ASSO HBW, and the manuscript language ought to be tempered accordingly.

L45-49. Break lengthy sentence into two parts.

L159-160 needs a reference.

L170. Preface sentence with "We hypothesize that..."

L202. Delete "investigation of"

Figure 1. I am not convinced by the authors response to my request and still believe that adding two contours - showing the max and min of the annual winter-maximum ice extent during the study period - would be very useful for helping the reader to orient to this environment.

Figure 2. Add a reference line for $y=0$ and a caption sentence explaining that the vertical lines demarcate seasons at the equinoxes and solstices (21st). The authors explained this to me but the information needs to be present in the manuscript.

Captions for Table 1 and Figure 3 please add text "...of the best fit model for acoustic presence of HBW at stations G3-4." NB if you remove the intercept from model (-1 into formula) this parameter should report as ONICPositive not (Intercept).

Supplement S2. Very small x-axes make it difficult to ascertain the seasons – could the boxplots or background be coloured by season? Please annotate what does a point (without boxplot) indicate e.g. a single daily observation that month.

Supplement S3 needs an explanation. Are these spatially showing the information from Supplement S2 or something slightly different? e.g why does Aug 2011 show data at G1 in S3 but not S2?

Reviewer #1 (Remarks to the Author):

The authors have dealt appropriately with my concerns expressed on their previous draft of this manuscript. I have no further comments.

Reviewer #2 (Remarks to the Author):

In general, I find the authors have done a commendable job incorporating most of my points and those of the other reviewer. This includes moderating the title, adding a new statistical analysis, clarifying the nature of HBW calling behaviour, and adding supplementary figures. Nevertheless, I do have some issues with how the new analysis was conducted and am still concerned that the interpretations regarding the role of sea ice (as raised previously) have not been sufficiently moderated. These two concerns are elaborated below, but I do believe that with another round of revision these issues can be addressed, and a very interesting paper published.

1. Statistical analysis

I am not clear why the (most abundant?) acoustic data from G1 were excluded. But, given that the analysis was applied to blended data from G3/G4 I am not sure that a mixed-effect model is necessary. If station is not required as a random effect – what random effect is included? – it may be better to just simplify to a GAM.

This is very pertinent, because the text indicates that through use of the `mgcv` `gamm` function a temporal correlation term (`corARMA` from the `nlme` package interfaced with `mgcv`) has been included. However, the `mgcv` package includes a clear warning on this:

WARNINGS... gamm performs poorly with binary data, since it uses PQL. It is better to use gam with s(...,bs="re") terms, or gamm4. gamm assumes that you know what you are doing!

The package indicates that the preferred (most numerically robust) approach for binomial GAMMs is via package `gamm4` (also from Simon Wood) which interfaces to `lme4`; noting that `nlme` correlation structures are not able to be included for these. From my reading however, this current study could avoid the dubious call to `gammPQL` (which provides only approximate MLEs) by simplifying to a GAM (which will also enable the `corARMA` correlation).

Reply - *We could not use data from all stations in a mixed model's framework (random effect: station ID) because:*

1) the number of stations is very small, beyond the recommendations of the minimum number of random effects to be used. I quote below a text from Ben Bolker's FAQs on GLMMs [<http://bbolker.github.io/mixedmodels-misc/glmmFAQ.html>]: "One point of particular relevance to 'modern' mixed model estimation (rather than 'classical' method-of-moments estimation) is that, for practical purposes, there must be a reasonable number of random-effects levels (e.g., blocks) – more than 5 or 6 at a minimum. ... "

2) The temporal coverage at stations is inconsistent, especially because not all stations recorded during the years 2015 and 2016. This is visible in Figure S2 and now also in Table 2:

Table 1. Information on passive acoustic recordings included in this study. The different recording periods at the five mooring positions were covered by different SonoVault recording units. For reference to earlier publications, the original mooring ID is listed in brackets.

Mooring ID	Latitude	Longitude	Recorder ID	Sampling Frequency (Hz)	Deployment Depth (m)	Recording Period
G1 (AWI227)	59 2.82 °S	000 5.78 °E	SV0002	5333	1007	2010-12-11 - 2011-05-21 2011-05-30
			SV1025	5333	1020	2011-06-14 - 2011-08-22
			SV1004	6857	1070	2012-12-11 - 2013-07-13 2016-12-22 - 2018-09-18
G2 (AWI229)	63 59.85 °S	000 1.84 °E	SV1000	5333	1007	2010-12-15 - 2011-06-18
			SV1010	5333	998	2012-12-14 - 2013-08-02
			SV1057	6857	970	2014-12-16 - 2016-05-19
G3 (AWI230)	66 2.01 °S	000 3.12 °E	SV1001	5333	934	2010-12-16 - 2012-04-13
			SV1009	5333	949	2012-05-06 - 2012-09-17 2013-01-07 - 2013-09-27
G4 (AWI231)	66 30.71 °S	000 1.51 °E	SV1002	5333	1083	2010-12-17 - 2012-02-05 2012-02-28 - 2012-07-30
			SV1058	6857	973	2012-08-04 - 2012-08-09 2012-08-11 - 2012-08-14
			SV1023	6857	859	2014-12-18 - 2016-05-28 2016-12-26 - 2018-10-28
G5 (AWI232)	68 59.94 °S	000 4.38 °E	SV1003	5333	987	2010-12-18 - 2012-05-09 2012-06-01 - 2012-08-10
			SV1011	5333	958	2012-12-17 - 2013-05-28 2013-06-19 - 2013-11-13
			SV1059	6857	999	2015-01-08 - 2015-01-26 2015-02-14 - 2015-02-21 2015-03-04 - 2015-08-24

We avoided gathering daily detection/non-detection data of all stations together, as we considered stations spatially-independent from one another (i.e., with large distance in between). This is true except for stations G3/G4, which are less than 50km apart, so pooling of these data seemed reasonable.

Thus, we do not have a random effect to be explicitly used in the model. However, the use of the usual 'mgcv::gam()' function in this study can lead to misleading conclusions as we know that observations (here, daily detections/no-detections) are temporally non-independent. Without considering temporal autocorrelation in the models, all of the gam effects are always highly significant (all very low p-values, even for SAM), only due to temporal autocorrelation (non-independence of the data, which violates the model assumptions; see representative figures of exemplary model without considering temporal autocorrelation below).

We considered temporal autocorrelation in the models by incorporating ARI (autoregressive correlation) and ARMA (autoregressive moving average) correlation structures, as recommended by Zuur et al. (2009), and we found ARMA to give better results and thus we used this in the final models.

To consider temporal autocorrelation, we cannot use `mgcv::gam()`, which does not support ARI and ARMA correlation structures. So, we did not use the '`mgcv::gamm()`' function for a random effect model per se., but to be able to consider temporal autocorrelation (using the correlation argument).

We acknowledge that '`gamm4::gamm4()`' function is more robust numerically than '`mgcv::gamm()`', especially for binary data (by avoiding the use of PQL). However, the current implementation of `gamm4` function does not support autocorrelated errors. In other words, it cannot handle temporal autocorrelation. Although `gamm4` is more numerically robust, our use of it without considering temporal autocorrelation can lead to misleading conclusions.

2. why G1 was dropped from the analysis if this station seems to have the most abundant data? I would suggest repeating the same model (separately) for G1. If indeed this station shows no climate effects, this is very interesting and suggests whales still migrate past ASSO but do not enter.

Reply – Although G1 is generally the location where most humpback whale vocal activity is recorded, G1 was not included in the statistical model because this recording location did not record during the years 2015 and 2016. This is visible in the supplementary material S2, but now we also added the recording dates for each station to Table 2, so this information is readily available to the reader. The reason we combined G3 and G4 for the statistical analysis was that we could get the best temporal coverage from the available data.

3. why it was necessary to explore categorical versions of the SAM and ONI covariates, rather than just use the continuous variables.

Reply – *With testing for the effect of both continuous and categorical versions of SAM and ONI, we wanted to include the possibility that the ecosystem and humpback whales might not directly respond to small-scale changes of SAM and ONI (i.e., represented by continuous variables), but might respond to large-scale changes from negative to positive phases or vice versa (i.e., represented by categorical variables). And this seems to be the case for at least ONI because humpback whales visit the area in summer independent of if ONI is in the negative or neutral phase, but do not seem to migrate into the ASSO when ONI is in its positive phase.*

4. is month necessary to include in the model if SIC essentially reflects season? Aren't these two covariates highly correlated?

Reply - *There is a clear monthly trend of SIC data. However, to include the annual cyclic trend of sea ice into the model, we need to include month as predictor. 'Month' was used in the model as cyclic regression spline 's(Month, bs = "cc")' to account for the cyclic nature and seasonal fluctuation of the data; i.e., a smooth transition between the end of the year (December) and the start of the next year (January). Please check the smooth transition at both ends for the response curve of 'Month' in Figure 3.*

5. the SAM term is relatively weak (p-value c.f. other terms) and looks at risk of being overfitted. To defend its inclusion authors might consider using the dredge function from the package MuMIn to evaluate all term combinations and use model ranking.

Reply – *We agree that SAM is at a critical position in the model. We tried implementing the 'MuMIn::dredge()' function by first refitting the model using the 'MuMIn::uGamm()' function, as required to enable using the dredge function with gamm models. However, dredge results were not very convincing. 'MuMIn::dredge()' considers a model only including SAM and month as the best model based on AIC. We believe that (again) the high temporal autocorrelation in our time-series data is responsible for the unexpected results of 'MuMIn::dredge()'. We estimated the best values for p and q (coefficients for the 'nlme::corARMA()' correlation function) using the full set of explanatory variables, which also have to be provided to 'MuMIn::dredge()'. However, the temporal autocorrelation of the residuals of nested models, e.g., by dropping SAM or Month, may not be best described with the same values of p and q, which in turn leads to the unconvincing results of 'MuMIn::dredge()'. Additionally, the different temporal resolutions of our predictors (i.e., daily vs. monthly) might also result in a biased selection by the dredge function. As an alternative, we reran our models, allowing the corARMA function to estimate its parameters (values of the autoregressive and moving average parameters) directly from our data instead of providing initial values (estimated by the forecast::auto.arima() function). This process takes much longer (of the order of days for each model) and has a much higher risk of failure (due to convergence issues), why we hesitated to try this approach before. Now, we took the time to rerun all our models with this approach and compared the converged models by their AIC. Indeed, SAM often turns out insignificant, depending on the setup of the correlation function, and in the two models with the lowest AICs, SAM turned out*

non-significant. Therefore, we replaced the old model in the manuscript with the resulting model with the lowest AIC, where SAM is non-significant, and adapted Table 1, Figure 3 and the corresponding text in the results and the methods.

Results: Modeling the effect of SIC, SAM, and ONI on the acoustic presence of humpback whales at the study location revealed that mainly SIC and ONI explain the observed pattern of humpback whale acoustic presence in the ASSO. The smoothed effects of month and SIC were highly significant because these variables explain seasonality in humpback whale presence on the feeding ground (Table 1, Figure 3; (Bombosch et al., 2014; Van Opzeeland et al., 2013)) in five out of seven years. The model showed that the ONI in the positive phase predicts a significantly lower probability of humpback whale acoustic presence than ONI in neutral or negative phases (Table 1, Figure 3). The smoothed effect of the SAM index was not statistically significant (Table 1, Figure 3). The model prediction for the SAM index showed lower predicted values at negative and high positive index values, although with higher uncertainties (Figure 3). This appears reasonable when looking at the original time-series. The negative phases of SAM were usually registered during winter when acoustic presences are naturally low, and extreme positive phases were only registered during summer 2015 and 2016 (Figure 2). Uncertainties at extreme index values (also for ONI) are high because these values are rare in the analyzed time-series, which potentially also explains the resulting non-significant effect of SAM. To quantify the relationship between humpback whale presence and climate indices with higher certainty, much longer time-series than presented here would be required.

Table 2. Summary of the best-fit model for the acoustic presence of humpback whales at stations G3/G4, including sea ice concentrations (SIC), SAM, and month as smooth terms, as well as ONI as a categorical predictor. Note that the factor levels of ONI as a categorical predictor are listed under the parametric coefficients.

Formula: $PA \sim s(SIC) + ONI + s(SAM) + s(Month)$				
Parametric coefficients:				
	Estimate	Std. Error	t value	Pr(> t)
ONI _{Positive}	- 5.8490	0.9888	- 5.915	3.87e-09 ***
ONI _{Positive} - ONI _{Neutral}	3.6750	0.9682	3.796	0.000151 ***
ONI _{Positive} - ONI _{Negative}	3.7672	1.0207	3.691	0.000229 ***
Approximate significance of smooth terms:				
	edf	Ref.df	F	p-value
s(SIC)	3.381	3.381	9.576	1.34e-06 ***
s(SAM)	2.103	2.103	1.561	0.167

s(Month)	4.635	8.000	4.387	7.23e-07 ***
R-sq.(adj) = 0.485				

Figure 3. Model predictions of the best-fit model for the acoustic presence of humpback whales at stations G3/G4, including the smooth terms sea ice concentration (SIC), month, and SAM as well as ONI as a categorical predictor (see methods for further explanation of categories). Gray-shaded areas in line plots depict 95%-confidence intervals.

Methods: Binomial GAMMs were applied to model the daily acoustic presence/absence of humpback whales at G3/4 as a function of Month, SIC, ONI (either continuous or categorical variable), and SAM (either continuous or categorical variable), including a model to account for temporal autocorrelation (functions *gamm* of the package *mgcv* (Wood, 2017) and *corARMA* of the package *nlme* (Pinheiro et al., 2020) for an autoregressive moving average (ARMA) model for the residuals). The optimal setup of starting values and orders for the implemented correlation structure was estimated in two ways: (1) with the function *auto.arima* (package *forecast* (Hyndman et al., 2020)), (2) by allowing the *corARMA* function to estimate its parameters directly from our data. Model selection was performed using the Akaike Information Criterion (AIC), adjusted r-squared values, and the analyses of residuals.

6. Interpretation regarding the role and dynamics of ASSO sea ice.

There are still several statements in the manuscript that need to be moderated or deleted:

L107-109. “Sea ice concentration during 2015 and 2016 did not show strong anomalies in the area of the Greenwich Meridian (Figure 2A).”

L140-144. “The major inter-annual variabilities in the acoustic presence of humpback whales on a Southern Ocean feeding ground seem to be driven by large-scale climate variabilities, but not local sea ice conditions. Our new data indicate that the local sea ice concentration is

not the main environmental factor explaining the presence of humpback whales in the Southern Ocean”.

L146-148. “Furthermore, the years 2015 and 2016 did not show strong abnormalities in the local sea ice concentrations at the Greenwich Meridian, which suggests that sea ice concentration was not the driver for the humpback whale acoustic absence in these years”.

As requested in the previous revision round, please reword this interpretation. It is not correct as currently phrased. There are no calculations in this manuscript pertaining to sea ice anomalies (at stations, or more broadly across the ASSO), so these statements are unsubstantiated (i.e. simply based on eyeballing Fig 2 and S3). In fact, the sea ice arrival looks very late in 2016. It is quite possible that the interannual climate oscillations (SAM/ONI) drive changes in ASSO ice extent, timing and duration; but this is not assessed here. In fact, authors state later in the paper: L156. “During positive values of SAM, the oceanic feeding areas of humpback whales in the ASSO show signs of reduced sea ice extent...” Exactly – these dynamics are probably linked – but you cannot say this in the Discussion after what you have said before!

I would suggest reformulating this interpretation along the lines of 1) SIC clearly plays a role in the intra-annual timing of whale presence (acoustic data is consistent with the expectation that presence begins to decline when sea ice concentration increases and that whales mainly disappear at sea ice concentrations above 80% (Figure 3)). 2) New finding: additionally, large-scale climate oscillations moderate the inter-annual presence of whales in the ASSO, going on to discuss what environmental changes these might drive (including ice dynamics). Statements about no abnormalities/anomalies need to be removed, as it is not assessed in the paper (unless the authors want to explore calculations of daily SIC anomalies relative to the climatological mean, at stations and/or across the ASSO).

Reply – *As recommended by the reviewer, we deleted one sentence in the results (“Sea ice concentration during 2015 and 2016 did not show strong anomalies in the area of the Greenwich Meridian (Figure 2A).”) and adjusted the first paragraph of the discussion accordingly:*

Local sea ice concentration is one of the most important environmental factors explaining the spatio-temporal distribution of baleen whales in the Southern Ocean (Bombosch et al., 2014; Filun et al., 2020; Širović et al., 2004; Thomisch et al., 2016). Similarly, sea ice dynamics play an important role in the intra-annual timing of humpback whale presence in the ASSO, showing that they move out of the area when SIC increases and that humpback whales are rarely present at SIC >80% (see also (Schall et al., 2020; Van Opzeeland et al., 2013)). Additionally, our new data indicate that large-scale climate variabilities drive the major inter-annual variability in the acoustic presence of humpback whales on a Southern Ocean feeding ground. The most likely pathway by which climate variabilities such as ENSO and SAM could affect humpback whale presence in the Southern Ocean is through their influence on Antarctic krill (*Euphausia superba*) (Loeb and Santora,

2015), since the availability and distribution of this primary prey species most likely is the main driver behind the spatio-temporal distribution of humpback whales in the ASSO.

7. Winter presence – this is still not evident up front in Fig 2A (to my eye) and while I appreciate the authors efforts to better include information on this in the new S2 and S3, unfortunately I still find myself squinting and wondering about better delivery of the data on year-round presence. A couple of suggestions might be compiling the 90+ maps in Supplementary S3 into a single animation file; developing a heatmap figure focusing on winter-spring months only (June-Nov) with station as column, month as row and heat colourmap indicating for example the number of days per month in which acoustic presence was detected; and/or a Table just documenting this information (for those years where Jun-Nov acoustic obs of HBW occurred). In any case these winter observations do seem to be extremely limited, indicating year-round presence to be the exception rather than the norm for ASSO HBW, and the manuscript language ought to be tempered accordingly.

Reply - *In accordance to the reviewer's suggestion, we replaced the maps in S3 with a heatmap showing the average number of days per month with humpback whale acoustic presence for all 5 stations. The heatmap also lists the actual number of days that HWs were on average present during month x at site x, making an additional table with these numbers obsolete. Furthermore, we agree with the reviewer that humpback whale winter acoustic presence is limited, but nevertheless seems to consistently occur over different sites and years of study. Please also refer to Schall et al. (2020) and Van Opzeeland et al. (2013) for further data from other years and recording sites. The fact that winter acoustic presence is not occasional, but occurs persistently in this study too, deserves explicit mention, as it once more illustrates the fact that humpback whales exhibit much more complex movement strategies than long assumed (see also Geijer et al., 2016). Nevertheless, we have tempered the statements in this context throughout the manuscript.*

Lines 119-121: Although humpback whale winter acoustic presence was limited compared to the summer proportions, the occurrence of calls in winter was persistent between years occurring at multiple sites (Supplementary material S3).

Lines 202-205: Similarly, sea ice dynamics play an important role for the intra-annual timing of humpback whale presence in the ASSO, showing that they move out of the area when sea ice concentration increases and that whales are rarely present at sea ice concentration >80% (see also Schall et al., 2020; Van Opzeeland et al., 2013).

Supplementary material S3:

Figure 7. Heatmap showing the average number of days with humpback whale acoustic presence for all months for recording stations G1-5. Darker colors indicate higher presence, numbers in cells represent the average number of days with presence per month per recording station.

8. L45-49. Break lengthy sentence into two parts.

Reply – *We divided the sentence into three separate sentences (lines 43-51):*

To improve the understanding of the ecological conditions under which humpback whales use the area as a feeding ground, we investigated the inter-annual changes in humpback whale acoustic presence in relation to three environmental parameters that are key to the Southern Ocean: 1) The Southern Annular Mode (SAM) which is the dominant pattern of natural climate variability in polar and subpolar regions of the Southern Hemisphere. 2) The El Niño Southern Oscillation (ENSO) causes periodic fluctuation of sea surface temperature and air pressure originating from the tropical Pacific. Both climate oscillations have large effects on the Southern Ocean productivity (Atkinson et al., 2019; Loeb et al., 2009; Loeb and Santora, 2015; Siegel, 2016).

9. L159-160 needs a reference.

Reply – *We added the following references in line 220-221:*

ENSO has the strongest effects on the Pacific sector of the Southern Ocean, including the Western Antarctic Peninsula (Siegel, 2016; Turner, 2004).

8 Siegel, V. *Biology and ecology of Antarctic krill*. (Springer, 2016).

35 Turner, J. The el nino–southern oscillation and antarctica. *International Journal of Climatology: A Journal of the Royal Meteorological Society* **24**, 1-31,

10. L170. Preface sentence with “We hypothesize that...”

Reply – *We adjusted the sentence accordingly (line 231):*

We therefore hypothesize that during the years 2015 and 2016, positive phases of both SAM and ENSO led to reduced densities of krill on the oceanic feeding grounds of humpback whales in the ASSO while potentially creating alternative prey resources in other areas.

11. L202. Delete “investigation of”

Reply – *We adjusted the sentence accordingly (line 283-286):*

Interannual trends in the distribution or health status (e.g.,(Bengtson Nash et al., 2018)) of humpback whales and other baleen whales from the South Atlantic, but also other areas, warrant further investigation to provide information to whale stock and fishery management.

12. Figure 1. I am not convinced by the authors response to my request and still believe that adding two contours - showing the max and min of the annual winter-maximum ice extent during the study period - would be very useful for helping the reader to orient to this environment.

Reply – *As requested by the reviewer, we added two lines to Figure 1 which represent the minimum and the maximum of the yearly maximum sea ice extent during the winters (21 June – 21 September) of 2011-2018.*

Figure 1. Bathymetric map of the Atlantic sector of the Southern Ocean (ASSO) including the geographic positions of the HAFOS (Hybrid Antarctic Float Observation System) mooring network in the ASSO (coastline and bathymetry data were obtained from (Amante and Eakins, 2009; Wessel and Smith, 1996)). The five mooring positions, G1-G5, visualized with colored dots (i.e., red, green, orange, yellow, magenta), represent the recording locations of the receivers (moored between 2010 and 2018) which were analyzed during this study. Positions G1-G5 form part of the HAFOS long-term mooring network (gray dots(Rettig et al., 2013)). The other recording positions (W6-13) were only active during 2013 and were therefore not included here (but see (Schall et al., 2020) for details). Light grey lines represent the minimum and maximum of the annual wintertime (21 June -21 September) maximum sea ice extent during the of the study period (2011-2018)^(Spreen et al., 2008). Please note, that the lines shown do not delineate the sea ice extent of the specific years with the maximum and minimum wintertime maximum sea ice extent, but - calculated independently for each longitude - the multi-year composite of the maximum and minimum of the wintertime maximum sea ice extent during this period.

13. Figure 2. Add a reference line for $y=0$ and a caption sentence explaining that the vertical lines demarcate seasons at the equinoxes and solstices (21st). The authors explained this to me but the information needs to be present in the manuscript.

Reply – We adjusted the figure and figure caption accordingly:

Figure 2. A: Average proportion of hours with humpback whale acoustic presence per month from the four oceanic recording locations (G1-G4) on the Greenwich Meridian from December 2010 until September 2018 (red bars). Gray-shaded areas represent months without recording data. The blue solid line and the right y-axis depict the daily averaged sea ice concentration within a 50 km radius around recording locations. **B:** Climatic variations from 2011 until 2018 indicated by three-month running means of the Southern Annular Mode index (SAM) as a dominant pattern of natural climate variability in polar and subpolar regions of the Southern Hemisphere and the Oceanic Niño Index (ONI) representing the periodic fluctuation of sea surface temperature and air pressure originating from the tropical Pacific. Time span of strong El Niño phase in 2015/16 is indicated by the yellow rectangle. Vertical gray lines indicate the onset of summer (S) and winter (W) and vertical dotted lines indicate the onset of spring and autumn (based on equinoxes and solstices). Horizontal dashed line represents zero-orientation line.

14. Captions for Table 1 and Figure 3 please add text "...of the best fit model for acoustic presence of HBW at stations G3-4." NB if you remove the intercept from model (-1 into formula) this parameter should report as ONIcPositive not (Intercept).

Reply – We added the requested information to the captions for Table 1 and Figure 3. In our models, we did not remove the intercept. The intercept of the parametric coefficient (ONI) represents the reference level used ($ONI_{Positive}$), and the other two lines compare the effect of the other two levels ($ONI_{Neutral}$ and $ONI_{Negative}$) to the reference level ($ONI_{Positive}$). We make this clear in the updated table (see reply to comment Nr. 5).

15. Supplement S2. Very small x-axes make it difficult to ascertain the seasons – could the boxplots or background be coloured by season? Please annotate what does a point (without boxplot) indicate e.g. a single daily observation that month.

Reply – We added background colors according to the seasons to the boxplot and added the requested information to the figure caption:

Figure 2. Boxplots of daily proportions of hours with humpback whale acoustic presence from the five recording positions (G1-G5) on the Greenwich Meridian displayed per month from December 2010 until September 2018 (center line, median; box limits, upper and lower quartiles; whiskers, 1.5x interquartile range; points, outliers). Grey bars represent months without recording data, yellow, red, blue and green shades indicate summer, fall, winter and spring seasons, respectively. Single points indicate single daily observations of humpback whale acoustic presence.

16. Supplement S3 needs an explanation. Are these spatially showing the information from

Supplement S2 or something slightly different? e.g why does Aug 2011 show data at G1 in S3 but not S2?

Reply – We added a figure legend to S3 to explain any unclarities (see below). S2 also shows humpback whale acoustic presence in August 2011 at G1:

Figure 7. Percentage of acoustic presence of humpback whales in the ASSO averaged per recording location and month for the years 2010-2018 (one year per page). Size and color of dots indicates a respective range of percentage of hours per month with humpback whale acoustic presence. The monthly averaged sea ice concentrations are depicted at a 25x25km resolution.

Literature

- Amante C, Eakins BW. 2009. ETOPO1 arc-minute global relief model: procedures, data sources and analysis.
- Atkinson A, Hill SL, Pakhomov EA, Siegel V, Reiss CS, Loeb VJ, Steinberg DK, Schmidt K, Tarling GA, Gerrish L, Sailley SF. 2019. Krill (*Euphausia superba*) distribution contracts southward during rapid regional warming. *Nat Clim Change*. 9:142-147.
- Bengtson Nash SM, Castrillon J, Eisenmann P, Fry B, Shuker JD, Cropp RA, Dawson A, Bignert A, Bohlin Nizzetto P, Waugh CA. 2018. Signals from the south; humpback whales carry messages of Antarctic sea ice ecosystem variability. *Global Change Biol*. 24:1500-1510.

- Bombosch A, Zitterbart DP, Van Opzeeland I, Frickenhaus S, Burkhardt E, Wisz MS, Boebel O. 2014. Predictive habitat modelling of humpback (*Megaptera novaeangliae*) and Antarctic minke (*Balaenoptera bonaerensis*) whales in the Southern Ocean as a planning tool for seismic surveys. *Deep-Sea Research Part I: Oceanographic Research Papers*. 91:101-114.
- Filun D, Thomisch K, Boebel O, Brey T, Širović A, Spiesecke S, Van Opzeeland I. 2020. Frozen verses: Antarctic minke whales (*Balaenoptera bonaerensis*) call predominantly during austral winter. *Royal Society Open Science*. 7:192112.
- Hyndman R, Athanasopoulos G, Bergmeir C, Caceres G, Chhay L, O'Hara-Wild M, Petropoulos F, Razbash S, Wang E, Yasmeeen F. 2020. `_forecast: Forecasting functions for time series and linear models_`. R package version 8.11.
- Loeb VJ, Hofmann EE, Klinck JM, Holm-Hansen O, White WB. 2009. ENSO and variability of the Antarctic Peninsula pelagic marine ecosystem. *Antarctic Science*. 21.
- Loeb VJ, Santora JA. 2015. Climate variability and spatiotemporal dynamics of five Southern Ocean krill species. *Progress in Oceanography*. 134:93-122.
- Pinheiro J, Bates D, DebRoy S, Sarkar D. 2020. `_nlme: Linear and nonlinear mixed effects models`. R package version 3.1-145. R CoreTeam.
- Rettig S, Boebel O, Menze S, Kindermann L, Thomisch K, van Opzeeland I. Local to basin scale arrays for passive acoustic monitoring in the Atlantic sector of the Southern Ocean. 1st International Conference and Exhibition on Underwater Acoustics2013; Corfu Island, Greece. p. 1669-1674.
- Schall E, Thomisch K, Boebel O, Gerlach G, Spiesecke S, Van Opzeeland I. 2020. Large-scale spatial variabilities in the humpback whale acoustic presence in the Atlantic sector of the Southern Ocean. *Royal Society Open Science*. 7:201347.
- Siegel V. 2016. *Biology and ecology of Antarctic krill*: Springer.
- Širović A, Hildebrand JA, Wiggins SM, McDonald MA, Moore SE, Thiele D. 2004. Seasonality of blue and fin whale calls and the influence of sea ice in the Western Antarctic Peninsula. *Deep Sea Research Part II: Topical Studies in Oceanography*. 51:2327-2344.
- Spreen G, Kaleschke L, Heygster G. 2008. Sea ice remote sensing using AMSR-E 89-GHz channels. *J Geophys Res-Oceans*. 113, C02S03.
- Thomisch K, Boebel O, Clark CW, Hagen W, Spiesecke S, Zitterbart DP, Van Opzeeland I. 2016. Spatio-temporal patterns in acoustic presence and distribution of Antarctic blue whales *Balaenoptera musculus intermedia* in the Weddell Sea. *Endangered Species Research*. 30:239-253.
- Turner J. 2004. The el nino–southern oscillation and antarctica. *International Journal of Climatology: A Journal of the Royal Meteorological Society*. 24:1-31.
- Van Opzeeland I, Van Parijs S, Kindermann L, Burkhardt E, Boebel O. 2013. Calling in the cold: pervasive acoustic presence of humpback whales (*Megaptera novaeangliae*) in Antarctic coastal waters. *PLoS One*. 8:1-7.
- Wessel P, Smith WH. 1996. A global, self-consistent, hierarchical, high-resolution shoreline database. *Journal of Geophysical Research: Solid Earth*. 101:8741-8743.
- Wood SN. 2017. *Generalized additive models: an introduction with R*: CRC press.
- Zuur A, Ieno EN, Walker N, Saveliev AA, Smith GM. 2009. *Mixed effects models and extensions in ecology with R*: Springer Science & Business Media.

Reviewers' Comments:

Reviewer #3:

Remarks to the Author:

I greatly appreciate the diligence with which the authors have approached all my comments. Indeed, I commend how honestly they have responded and updated their results. While modelling this type of data is very challenging, I am satisfied by the changes made and do not wish to delay further the publication of this very interesting manuscript. I have a couple of minor points below, and a note for the authors for the future, but I do not need to see this manuscript again. My congratulations and best wishes.

1. Responses to comments No. 3 and 4. The authors explain well to me their reasoning but I suggest a sentence on each should be included in the methods - i.e. regarding the categorical term, replacing the repeated phrase "either continuous or categorical variable" with a short reasoning. Similarly, include a short explanatory sentence about the SIC/Month and use of a cyclic term.

2. The additional information included in Table 2 on Recording period is very helpful – could the actual number of days data recorded be included in parenthesis for each period so the reader doesn't have to mentally calculate? e.g.

2010-12-15 - 2011-06-18 (185)

2012-12-14 - 2013-08-02 (231)

2014-12-16 - 2016-05-19 (520, or less, if there are some few missing days in the interim etc.)

3. On the use of corARMA() new lines 294-298; here or somewhere in the MS should specify that the final model reported had these parameters estimated (as explained in reply to comment No 5.)

For the authors' future endeavors: on further investigation it looks as if mgcv:gam() accepts but essentially ignores inclusion of a correlation = corAR1() or corARMA() term! Ugh. Other feasible options worth exploring might be:

- using gamm4 including a time variable in the specification of random effects, nesting year within site e.g. random= \sim (time|Site/Year) (five sites with multiple years is a lot of data, and mixed models should be capable of handling unbalanced data wrt temporal coverage)

-switching to the glmmTMB package which enables use of a corAR1() structure with binomial data

Best of luck!